# Can Group Exercise Programs Improve Health Outcomes in Pregnant Women? An Updated Systematic Review

**DOI:** 10.3390/ijerph19084875

**Published:** 2022-04-17

**Authors:** Rebeca de Castro, Raul Antunes, Diogo Mendes, Anna Szumilewicz, Rita Santos-Rocha

**Affiliations:** 1ESECS—Polytechnic Institute of Leiria, 2411-901 Leiria, Portugal; rebecabettencourt@gmail.com (R.d.C.); diogo.l.mendes@ipleiria.pt (D.M.); 2Life Quality Research Centre (CIEQV), 2040-413 Leiria, Portugal; 3Center for Innovative Care and Health Technology (ciTechCare), Polytechnic of Leiria, 2410-541 Leiria, Portugal; 4Department of Fitness, Faculty of Physical Culture, Gdansk University of Physical Education and Sport, 80-336 Gdansk, Poland; anna.szumilewicz@awf.gda.pl; 5ESDRM Sport Sciences School of Rio Maior—Polytechnic Institute of Santarém, 2040-413 Rio Maior, Portugal; ritasantosrocha@esdrm.ipsantarem.pt; 6CIPER Interdisciplinary Centre for the Study of Human Performance, Faculty of Human Kinetics (FMH), University of Lisbon, 1499-002 Cruz Quebrada, Portugal

**Keywords:** physical activity, exercise, health, fitness, pregnancy, postpartum

## Abstract

Current scientific evidence supports the recommendation to initiate or continue physical exercise in healthy pregnant women. Group exercise programs have positive effects on improving health, well-being, and social support. In 2015, a systematic review was provided to evaluate the evidence on the effectiveness of group exercise programs in improving pregnant women’s and newborns’ health outcomes and to assess the content of the programs. This review aims to update this knowledge between 2015 and 2020. The exercise program designs were analyzed with the Consensus of Exercise Reporting Template (CERT) model, the compliance with the current guidelines, and effectiveness in the maternal health and fitness parameters. Three databases were used to conduct literature searches. Thirty-one randomized control trials were selected for analysis. All studies followed a supervised group exercise program including aerobic, resistance, pelvic floor training, stretching, and relaxation sections. Group interventions during pregnancy improved health and fitness outcomes for the women and newborns, although some gaps were identified in the interventions. Multidisciplinary teams of exercise and health professionals should advise pregnant women that group exercise improves a wide range of health outcomes for them and their newborns.

## 1. Introduction

Regular physical activity in all stages of life, including pregnancy, promotes health benefits, such as maintaining and improving cardiorespiratory fitness and reducing the risk of obesity and associated comorbidities [1,2]. Pregnancy is a special stage where lifestyle behaviors, including physical activity, can significantly affect maternal [1,2] and fetal health [3]. The American College of Obstetricians and Gynaecologists (ACOG) reports that women who begin their pregnancy without a healthy lifestyle should be encouraged to adopt it in pre-pregnancy and pregnancy periods and should follow a more gradual progression of exercise intensity [2]. For pregnant women who want to adopt the practice of physical exercise during pregnancy, exercise intensity should be adjusted according to their physical activity and exercise levels. Moderate-intensity physical exercise has the most scientific evidence and the strongest recommendation. These guidelines suggest that pregnant women without contraindications should accumulate at least 150 min/week of moderate-intensity aerobic exercise spread across, at least, 3 days per week, in sessions of 30–60 min [2]. The practice can start in the first trimester of gestation and can be maintained until delivery (as tolerated). They should incorporate a variety of aerobic exercises, resistance training, yoga, and stretching. Some examples of safe aerobic activities to perform during pregnancy are walking, stationary cycling, dancing, aerobics, step, and water exercises [1,2]. The strength training consists of low resistance exercises with barbells, dumbbells, and resistance bands [4,5]. Pelvic Floor Muscle Training (PFMT) should be performed to reduce the odds of urinary incontinence (UI) and other pelvic floor dysfunctions [6].

Pregnant women who performed regular high-intensity exercise before pregnancy and who have an uncomplicated and healthy pregnancy should be able to maintain the practice of high-intensity exercise; while, being flexible for possible adjustments according to their condition or type of exercise [2]. Positive outcomes of regular exercise (i.e., aerobic dance) were also obtained in healthy former inactive pregnant women [7].

The Canadian Guidelines present maternal heart rate as a measure of exercise intensity [1]. The ACOG Guidelines refuted it, outlining that due to blunted heart rate responses to exercise reported in pregnant women, the use of ratings of perceived exertion may be a more effective means to monitor exercise intensity. The 6–20 Borg’s Scale can be used, between 13–14 values for moderate intensity. The talk test is another valid way of gauging exercise intensity [2]. These recommendations are intended for pregnant women without contraindications for physical exercise practice.

In 2014, Evenson et al. [8] compared pregnancy-related physical activity guidelines from around the world. The authors suggested that there should be more consensus on the guidelines to enable proper cooperation between health professionals, exercise professionals, and pregnant women [8]. This review supported the creation of a common language to facilitate communication between the various stakeholders, which may help increase the adherence to the physical activity guidelines by both the providers and their patients. More recently, these authors fostered that the four recent guidelines can facilitate the use of updated recommendations by healthcare providers regarding physical activity during pregnancy [9]. Furthermore, to facilitate the use of guidelines in practice, there are other tools available at the ACOG and the Canadian Society for Exercise Physiology (CSEP) websites, and specific guidelines for exercise prescription and selection should be followed [10].

The ACOG guidelines support that exercise during pregnancy can: control the gestational weight gain, gestational diabetes mellitus, and gestational hypertensive disorders; increase the vaginal delivery probability; and control the birth weight. The evidence reported a 50% and 35% reduction in prenatal and postnatal urinary incontinence, respectively. However, the recommendation level is weak with low-quality evidence [2].

Despite the benefits of exercise in pregnancy, supported by the most updated guidelines [1,2,11], the physical exercise program adherence revealed itself as a limitation and as a significant challenge in many randomized control trials (RCT) [12]. Nevertheless, the loss to follow-up seems to be higher and adherence to the intervention was lower for unsupervised versus supervised group exercise [13]. On the contrary, almost the total of participants of the group exercise-based RCT of Haakstad et al. [14] answered that they would recommend this type of exercise for pregnant friends. Reported motives and health benefits included better aerobic capacity, increased energy levels, and exercise enjoyment [14].

In 2015, a systematic review was provided to evaluate the available evidence on the effectiveness of group exercise programs in improving women’s health outcomes during pregnancy, as well as newborn’s health outcomes, and to assess the content of the programs [15]. The present systematic review aims to update the knowledge about the effectiveness of exercise in a group context during pregnancy, between 2015 and 2020. Additionally, given the lack of consensus between the guidelines, it was pertinent to contrast the RCTs, which are presented as the evidence base for guidelines. The authors analyzed the physical exercise program design, consistency with the guidelines, and identifyied limitations in the interventions that can be improved in the future.

## 2. Materials and Methods

The authors followed the methodological approach described by the Preferred Reporting Items for Systematic Reviews and Meta-Analyses (PRISMA 2020). The PRISMA Statement consists of a 27-item checklist and a four-phase flow diagram, and it is aimed at helping authors improve the reporting of systematic reviews.

### 2.1. Criteria for Considering Studies for This Review

#### 2.1.1. Types of Studies

This review included RCT studies published in the English, Portuguese, and Spanish languages, between 2015 and 2020, to find the most recent data.

#### 2.1.2. Participants

This review included studies that recruited adult pregnant women without known pregnancy-related medical conditions.

#### 2.1.3. Type of Interventions

The articles included in this review were based on prenatal in-person group exercise interventions with a minimum duration of 12 weeks, supervised by physical exercise specialists or exercise physiologists. Interventions designed explicitly for underweight, overweight, and obese pregnant women, or other pregnancy-related medical conditions were excluded.

The comparator criteria were: (1) no exercise control group (i.e., receiving “standard care“, or included in a “waiting list”); (2) no exercise but with other types of intervention groups (e.g., motivational counseling); (3) intervention group 2, i.e., control group engaged in a different exercise program (e.g., supervised vs. non-supervised program, group vs. personal training, multicomponent vs. not mixed exercise, etc.).

#### 2.1.4. Outcomes Measures

Considering the aims of this systematic review, the main outcome variables in the analysis are related to maternal health (i.e., quality of life, weight gain, gestational diabetes, cholesterol, hypertension, pre-eclampsia, well-being, depression, rest heart rate, sleep quality, low back pain), maternal physical activity (i.e., level and type of physical activity), and maternal fitness (i.e., cardiorespiratory fitness, strength, flexibility, balance, coordination, pelvic floor muscle strength), and labor outcomes (i.e., mode of delivery, birth weight, fetal cardiac function, and Apgar score).

### 2.2. Search Strategy

Studies were identified by searching three databases: PubMed, Scopus, and Scielo, from October 2020 to January 2021. The following search terms were conjoined to be identified in the title or abstract of the article or the Medical Subject Headings terms: “pregnancy/gestation/pregnant”and “physical activity/exercise/group exercise”. PubMed allowed the search of RCT studies, “female”, and “humans”.

### 2.3. Study Selection

The study selection was conducted in three phases: the first phase was the screening of the articles by their titles against the inclusion criteria; the second phase was the screening of the abstracts, and; the third phase was the full screening of the article. During the first and second phases, in the case of doubts about including any article, the screening was delayed for the next phase. At this point, all team members participated in the discussion to resolve it. In the case of duplicated studies, the authors randomly picked one from the three databases.

### 2.4. Data Extraction

The following data was extracted from each selected article:
author and year of publication;sample size, characteristics of participants, country and location where the study was performed, and gestational week;purpose of study, study design including type, frequency, and duration of intervention, exercise intensity, equipment, exercise and health specialists leading the program, and the number of participants in each session;maternal and fetal health and fitness outcomes measured;results of outcomes measured.

### 2.5. Quality Assessment of the Studies

Version 2 of the Cochrane risk-of-bias tool for randomized trials (RoB 2) was used to assess the quality and risk of bias of each study, in order to reduce bias and check the internal and statistical validity of each study. It is structured in different domains focused on the study’s design, conduct, and analysis. Each domain has some questions that collect information relevant to the risk of bias. The proposed judgment on the risk of bias arising from each domain is generated by an algorithm based on the answers to the questions. The judgment may be ‘Low’ or ‘High’ risk of bias or express ‘Some concerns’. If the author disagrees with the algorithmic answer, the final judgment can be changed, always prevailing as the final decision. Two of the researchers independently assessed the risk of bias and the results were compared and a consensus was reached for each study.

### 2.6. The Consensus of Exercise Reporting Template (CERT)

In addition to the previous systematic review [15], this article evaluated intervention design and reported its limitations. The CERT model evaluated the physical exercise program design through key items considered essential to report the replicable capacity of intervention. The CERT model comprises 19 questions (refer to 16 items) organized in 7 sections/domains: what (materials and equipment); who (provider); how (delivery); where (location); when, how much (dosage); tailoring (what, how); and how well (compliance/planned and actual). The possible answers are “YES” and “NOT”. Each answer with “YES” offers 1 point to study. The studies with higher scores have higher quality interventions. The higher the score the more easily the intervention can be understood and replicated. No study was excluded by the CERT score [16].

## 3. Results

### 3.1. Study Selection

Initially, 580 articles were identified in the databases search. Sixteen articles duplicated were removed. Considering the inclusion and exclusion criteria, in the screening based on title or abstract, 484 articles were excluded. In the eligibility phase, 80 articles were assessed. After a full-text review of these articles, thirty-one were selected for analysis as shown in Figure 1. 

### 3.2. Characteristics of the Studies

Table 1 provides the quality assessment of the studies using the RoB2, as well as the total score of each study.

Table 2 provides the characteristics of the studies, which were described according to sample size, location, duration, aims of the study, type, and description of the intervention, primary and secondary outcome variables, and results.

This review contains 31 studies including a total of 7560 pregnant women. The sample size across studies ranged from 36 to 840 pregnant women. All studies present an RCT design. Concerning the span of intervention, most studies assessed the influence of a physical exercise program on the outcome variables, during the second and third trimesters of pregnancy but also during the entire pregnancy [4,5,7,12,17,18,19,20,21,22,23,24,25,26,27,28,29,30,31,32,33,34,35,36,37,38,39,40,41,42,43]. All interventions lasted at least 12 weeks. The studies were conducted in the following countries: Spain [4,5,17,20,21,26,29,30,31,32,33,34,38,40,41,42,43], Argentina [39], Norway [6,18,19,23,24,28,37], Brazil [12,22], Iran [25], USA [27], Egypt [35], India [36].

**Table 1 ijerph-19-04875-t001:** Quality assessment and RoB2 score.

Authors	D1	D2	D3	D4	D5	Overall
Sánchez-García, J. (2019) [43]	**+**	**+**	**+**	**+**	**+**	**+**
Cordero, J. (2017) [17]	**+**	**+**	**+**	**+**	**+**	**+**
Bacchi, M. (2017) [39]	**+**	**+**	**+**	**+**	**+**	**+**
Haakstad, L. (2016) [7]	**+**	**+**	**+**	**+**	**+**	**+**
Sagedal, L.R. (2017) [18]	**+**	**+**	**+**	**+**	**+**	**+**
Sagedal, L.R. (2017) [19]	**+**	**+**	**+**	**!**	**+**	**!**
A-Cordero, M. (2018) [20]	**+**	**+**	**+**	**+**	**+**	**+**
Barakat, R. (2018) [40]	**+**	**+**	**+**	**+**	**+**	**+**
Sanda, B. (2018) [37]	**!**	**+**	**!**	**+**	**+**	**!**
Terrones, M. (2018) [21]	**+**	**+**	**+**	**+**	**!**	**!**
Blanque, R. (2017) [38]	**+**	**+**	**+**	**+**	**+**	**+**
Dias, N. (2017) [22]	**+**	**+**	**+**	**+**	**+**	**+**
Haakstad, L. (2015) [23]	**+**	**+**	**!**	**+**	**+**	**!**
Gustafsson, M.K. (2015) [24]	**+**	**+**	**+**	**+**	**+**	**+**
Charkamyani, F. (2019) [25]	**+**	**+**	**+**	**+**	**+**	**+**
Perales, M. (2015) [26]	**+**	**+**	**+**	**+**	**+**	**+**
Palaez, M. (2019) [41]	**+**	**+**	**+**	**+**	**+**	**+**
Clark, E. (2019) [27]	**+**	**+**	**+**	**+**	**!**	**!**
Haakstad, L. 2020) [28]	**+**	**+**	**+**	**+**	**+**	**+**
Blanque, R. (2020) [29]	**+**	**+**	**+**	**+**	**+**	**+**
Blanque, R. (2020) [30]	**+**	**+**	**+**	**+**	**+**	**+**
Reoyo, O. (2019) [31]	**+**	**+**	**+**	**+**	**+**	**+**
Blanque, R. (2019) [32]	**+**	**+**	**+**	**+**	**+**	**+**
Brik, M. (2018) [33]	**+**	**+**	**+**	**+**	**+**	**+**
Coll, C. (2019) [12]	**+**	**+**	**+**	**+**	**+**	**+**
Blanque, R. (2019) [34]	**+**	**+**	**+**	**+**	**+**	**+**
Awad, E. (2019) [35]	**+**	**+**	**-**	**+**	**!**	**-**
Barakat, R. (2017) [5]	**+**	**+**	**+**	**+**	**+**	**+**
Pawalia, A. (2017) [36]	**−**	**!**	**!**	**+**	**+**	**−**
Barakat, R. (2016) [4]	**+**	**+**	**+**	**+**	**+**	**+**
Cordero, Y. (2015) [42]	**!**	**+**	**+**	**+**	**+**	**!**

Note 1: D1—Randomisation Process; D2—Deviations from the intended interventions; D3—Missing outcome data; D4—Measurement of the outcome; D5—Selection of the reported result; Low Risk (**+**); Some Concerns (**!**); High Risk (**−**).

**Table 2 ijerph-19-04875-t002:** Charactheristics of the Studies.

Authors and Year of Publication	Participants and Location	Objectives	Study Design	Measures	Results
Juan Carlos Sánchez-García et al. (2019) [43]	**N =** 129 IG = 65 + CG = 64 Healthy PW without contraindications for practice ACOG (2015). **Gestational Week:** 20th **Location:** Granada, Spain	To examine the gestational weight gain and postpartum on pregnant women who realized moderate physical exercise in an aquatic environment.	**Type of Intervention:** RCT (CG × IG) **Description:** The IG had access to a moderate intensity physical exercise program in water. The intervention was supervised by an exercise specialist. The CG received usual care and general advice on the benefits of exercise. **Intensity Measurement:** Borg Scale or Heart Rate **Time of session:** 60′ **Frequency:** 3x / week **Duration of Intervention:** 17 weeks	**Primary:** GWG, weight retention (16th and 18th postpartum week), and newborn weight. **Secondary:** The level of physical activity (GPAQ at 12th gestational week) and baseline maternal characteristics.	The study showed that the variables of GWG, weight retention (4 and 7 months), and newborn weight presented lower values on IG.
JAM Cordero et al. (2017) [17]	**N =** 140 IG = 70 + CG = 70 Healthy PW without contraindications for practice (ACOG 2015). **Gestational Week:** 20th **Location:** Granada, Spain	To examine the effect of a physical exercise program with moderate intensity during pregnancy in water on newborn weight.	**Type of Intervention:** RCT (CG × IG) **Description:** The IG had access to a moderate intensity physical exercise program in water. The intervention was supervised by an exercise specialist. The CG received usual care and general advice on the benefits of exercise. **Intensity Measurement:** Borg Scale or Heart Rate **Time of session:** 60′ **Frequency:** 3x / week **Duration of Intervention:** 17 weeks	**Primary:** Days of gestation and newborn weight. **Secondary:** maternal weight (1st and 3rd trimesters) and baseline maternal characteristics.	Lower values of newborn weight in IG. However, these differences did not represent clinical trends because both groups are in a normal state of weight.
Mariano Bacchi et al. (2017) [39]	**N =** 111 IG = 49 + CG = 62 Healthy PW without contraindications for practice (ACOG 2002). **Gestational Week:** 10th **Location:** Buenos Aires, Argentina.	To examine the influence of supervision and regular water activities program during pregnancy on maternal weight gain and birth weight.	**Type of Intervention:** RCT (CG × IG) **Description:** Moderate physical exercise intervention in water. Sessions included aerobic, strength, and aquatic activities, in standing, supine and ventral positions. Relax phase included relaxing, stretching, and breathing exercises. The intervention was supervised by an exercise specialist. The CG received usual care and did not report any physical exercise during pregnancy. Water Temperature: 28.5° to 29° **Intensity Measurement:** BORG Scale **Time of session:** 55 to 60 min **Frequency:** 3 per/week **Duration of Intervention:** 26 weeks	**Primary:** GWG and birth weight. **Secondary:** baseline maternal characteristics.	Higher percentage of women with excessive maternal weight gain in CG No differences between groups on the birth weight variable.
Lene A. H. Haakstad et al. (2015) [4]	**N =** 61 IG = 35 + CG = 26 Healthy PW without contraindications for practice (ACOG 2002). **Gestational Week:** 24th **Location:** Oslo, Norway.	To evaluate the effect of regular exercise on maternal arterial blood pressure at rest and during uphill walking.	**Type of Intervention:** RCT (CG × IG) **Description:** Moderate intensity physical exercise program. Each session included warm-up, aerobic, strength (general and pelvic), and relax phases. The participants were advised to do at least 30′ of moderate physical exercise on the rest of the days. The CG received usual care and recommendations for maintaining physical activity without receiving some incentive for physical exercise practice. **Intensity Measurement:** Borg Scale **Time of session:** 60 min **Frequency:** 2x/week **Duration of Intervention:** 12 weeks	**Primary:** Resting systolic and diastolic blood pressure (three times by auscultatory techniques) and walking systolic and diastolic blood pressure (during monitoring walking). This was performed before and after the intervention.	Lower values of blood pressure in rest in IG. The values of blood pressure during exercise were lower on IG.
Linda R. Sagedal et al. (2016) [18]	**N =** 591 IG = 296 + CG = 295 Healthy PW without contraindications for practice (ACOG 2002). **Gestational Week:** ≤20th **Location:** Kristiansand and Mandal, Norway.	To examine whether a lifestyle intervention on nutrition and physical exercise in pregnancy limits GWG and provides measurable health benefits for mother and newborn.	**Type of Intervention:** RCT (CG × IG) **Description:** Dietary and physical exercise intervention. The group was encouraged to engage in 30′ of moderate-intensity physical activity on three additional days per week. Counselling dietary was performed by phone. The sessions were supervised by the therapists or students in sports science, trained, and quality controlled by the team. The CG received routine prenatal care in accordance with Norwegian standards. They received a booklet with advice on prenatal nutrition and physical activity. **Intensity Measurement:** Borg Scale **Time of session:** 60 min **Frequency:** 2x/week **Duration:** 12 weeks	**Primary:** GWG, birthweight, the proportion of infants weighing >4000 g, and the incidence of operative deliveries. **Secondary:** the proportion of newborns of birthweight ≥90th percentile, the incidence of delivery complications.	Statistically significant decrease in GWG in the IG. The intervention did not decrease the incidence of pregnancy complications or operative delivery and had no effect on fetal weight or neonatal outcomes.
Linda R. Sagedal et al. (2017) [19]	**N =** 591 IG = 296 + CG = 295 Healthy PW without contraindications for practice (ACOG 2002). **Gestational Week:** ≤20th **Location:** Kristiansand and Mandal, Norway.	To examine the effect of the exercise intervention on glucose metabolism, including an assessment of the subgroups of normal-weight and overweight/obese participants	Intervention: RCT (CG × IG) **Description:** Dietary and physical exercise intervention. The group was encouraged to engage in 30′ of moderate-intensity physical activity on three additional days per week. Counselling dietary was performed by phone. The sessions were supervised by the therapists or students in sports science, trained, and quality controlled by the team. The CG received routine prenatal care in accordance with Norwegian standards. They received a booklet with advice on prenatal nutrition and physical activity. **Intensity Measurement:** Borg Scale **Time of session:** 60 min **Frequency:** 2x/week **Duration:** 12 weeks	**Primary:** GWG, birth weight of term infants, the proportion of term infants >4000 g, maternal fat percent at 36 gestational weeks, and the incidence of operative deliveries. **Secondary:** proportion of women with elevated 2-h glucose tolerance tests and measurement of hormones related to glucose metabolism.	Statistically significant reduction of GWG. In variables of infants, there were no differences. The levels of insulin were statistically significantly lower in IG. In the normal weight sub-group, there was a reduction in levels of insulin and leptin. In the obesity sub-group, the only difference was in fasted glucose values.
María José Aguilar-Cordero et al. (2019) [20]	**N =** 129 GI = 65 + GC = 64 Healthy PW without contraindications for practice (ACOG 2015). **Gestational Week:** 12th/20th **Location:** Granada, Spain.	To determine whether physical activity during pregnancy alleviates Postpartum Depression	**Type of Intervention:** RCT (CG × IG) **Description:** The IG had access to a moderate intensity physical exercise program in water. The intervention was supervised by an exercise specialist. The CG received usual care and general advice on the benefits of exercise. **Intensity Measurement:** Borg Scale or Heart Rate **Time of session:** 60′ **Frequency:** 3x/week **Duration of Intervention:** 17 weeks	**Primary:** Prevalence of PPD. **Secondary:** Baseline maternal characteristics and GWG.	In the normal weight category, for PPD evaluation there were statistically significant differences between groups. However, none of the groups were at high risk of postpartum depression. For overweight and obesity categories there were statistically significant differences. Contrary to a category of normal weight, the CG showed values of a high risk of postpartum depression. Lower weight gain in IG.
Ruben Barakat et al. (2018) [40]	**N =** 456 GI = 234 + GC = 222 Healthy PW without contraindications for practice (ACOG 2015). **Gestational Week:** 8th/10th **Location:** Madrid, Spain.	Examine the effects of an exercise program throughout pregnancy on maternal weight gain and the prevalence of gestational diabetes	**Type of Intervention:** RCT (CG × IG) **Description:** Moderate intensity physical exercise intervention. Each session included warm-up, aerobic, light muscle strengthening, coordination, balance, and stretching exercises. All sessions were accompanied by music and the room had appropriate conditions to practice (altitude 600 m; temperature 19–21 °C; humidity 50–60%). The CG attended regular scheduled visits to their obstetricians and midwives, usually every 4–5 weeks until the 36–38th week of gestation and then weekly until delivery. They received general nutrition and physical activity counseling from their healthcare provider. **Intensity Measurement:** Borg Scale (12–14) **Time of session:** 55 to 60 min **Frequency:** 3x/week **Duration:** 30 weeks	**Primary:** GWG, excessive gestational weight gain (yes/no), GDM, 1 h OGTT. **Secondary:** maternal gestational age at delivery, type of delivery, and birth weight	IG presented lower GWG compared with CG. The ratio of women that gained excessive weight was higher in the CG than in the IG. IG presented statistically significant lower values of OGTT results compared with CG. The ratio of women diagnosed with GDM was higher in the CG than the IG, with statistically significant differences. The results just showed that the ratio of neonate macrosomia was slightly higher in CG than in the IG.
Birgitte Sanda et al. (2018) [37]	**N =** 606 GI = 295 + GC = 294 Healthy PW without contraindications for practice (ACOG 2002). **Gestational Week:** 18th **Location:** Norway.	Examine the effect of a lifestyle intervention including group exercise classes, as well as the possible influence of physical activity level in late pregnancy, on labor outcomes.	**Type of Intervention:** RCT (CG × IG) **Description:** Dietary and physical activity intervention. The IG was encouraged to engage in 30′ of moderate-intensity physical activity on three additional days per week. Counselling dietary was performed by phone. The sessions were supervised by the therapists or students in sports science, trained, and quality controlled by the team. The CG received routine prenatal care in accordance with Norwegian standards. They also received a booklet with advice on prenatal nutrition and physical activity. **Intensity Measurement:** Borg Scale (12–14) **Time of session:** 60 min **Frequency:** 2 per/week **Duration:** 30 weeks	**Primary:** The duration and type of labor. **Secondary:** Baseline maternal characteristics, other information of labor.	IG experienced a longer 1st stage of labor with a statistically significant difference compared with CG. PW with a high level of PA levels in late pregnancy had lower odds for acute cesarean delivery compared to women with low levels. Epidural analgesia was more common among in the low active group compared to women in the high active group.
Marina Vargas-Terrones et al. (2018) [21]	**N =** 124 GI = 70 + GC = 54 Healthy PW without contraindications for practice (ACOG 2015). **Gestational Week:** <16th **Location:** Madrid, Spain.	To examine the effect of an exercise program during pregnancy on the risk of perinatal depression.	**Type of Intervention:** RCT (CG × IG) **Description:** Moderate intensity physical exercise intervention. Each session included warm-up, aerobic, strength exercises, and stretching and relaxation. The sessions were supervised by a qualified fitness specialist. Both groups received usual care from health professionals of the hospital and the general recommendations of nutrition and exercise. **Intensity Measurement:** Borg Scale (12–14) or heart rate (55–60%) **Time of session:** 60 min **Frequency:** 3 per/week **Duration:** 30 weeks	**Primary:** Risk of depression was measured at the beginning of the study (12–16 weeks), at gestational week 38, and the 6th week postpartum. **Secondary:** Baseline maternal characteristics.	The prevalence of women with depression was lower in IG. These differences were shown at gestational week 38 and week 6 postpartum.
R. Rodriguez Blanque et al. (2017) [38]	**N =** 134 GI = 67 + GC = 67 Healthy PW without contraindications for practice (ACOG 2015). **Gestational Week:** 12th/20th **Location:** Granada, Spain.	To determine whether, in pregnant women, there is an association between moderate-intensity physical activity in an aquatic environment and sleep quality.	**Type of Intervention:** RCT (CG × IG) **Description:** The IG had access to moderate intensity physical exercise program in water. The intervention was supervised by an exercise specialist. The CG received usual care and general advice on the benefits of exercise. **Intensity Measurement:** Borg Scale or Heart Rate **Time of session:** 60′ **Frequency:** 3x/week **Duration of Intervention:** 17 weeks	**Primary:** Sleep quality was evaluated in the 1st and 3rd trimesters. **Secondary:** Baseline maternal characteristics.	This study showed that IG had better results in quality, duration, latency, and regular efficiency of sleep, compared with CG.
Naiara T. Dias et al. (2017) [22]	**N =** 50 GI = 25 + GC = 25 Healthy PW without contraindications for practice (ACOG 2015). **Gestational Week:** 14th/16th **Location:** Uberlândia, Brazil.	Evaluate the effectiveness of a Pilates exercise program with PFM contraction compared to a conventional intervention in pregnant women.	**Intervention:** RCT (CG × IG) **Description:** Pilates exercise intervention. The intervention intensity increased after an adaptation period of 4 weeks. Mats, therapeutic balls, and elastic bands were used. The intervention was supervised by an exercise specialist. The CG underwent walking for 10′ and strengthening exercises of the lower libs, upper limbs, and trunk with an elastic band and body weight resistance. At the end of each session, the women performed stretching and relaxation exercises. No type of instruction or verbal command was given regarding the PFM and abdominal muscle contraction. **Intensity Measurement:** Borg Scale (13–14) **Time of session:** 60 min **Frequency:** 2x/week **Duration:** 18 weeks	**Primary:** PFM strength (manometer) measured at 14th and 16th weeks and, again between the 32nd and 34th weeks of gestation. **Secondary:** digital palpation variables—PFM strength using Oxford Scale, PFM endurance, and PFM repeatability.	There were no significant differences between groups for the PFM strength assessed by manometer. The IG presented significantly better results on PFM strength measured through the Oxford scale, PFM endurance, and PFM repeatability, compared to CG.
Lene A.H. Haakstad et al. (2016) [23]	**N =** 105 GI = 52 + GC = 53 Healthy PW without contraindications for practice (ACOG 2002). **Gestational Week:** <24th **Location:** Oslo, Norway.	To examine the effects of supervised group exercise on maternal psychological outcomes and commonly reported pregnancy complaints.	**Type of Intervention:** RCT (CG × IG) **Description:** Moderate intensity physical exercise intervention. Each session included warm-up, cardiovascular training, strength (core muscles), and stretch and relaxation phase. The intervention was supervised by an exercise specialist. The CG was asked to continue their usual physical activity habits and were neither encouraged nor discouraged from exercising. **Intensity Measurement:** Borg Scale (12–14) **Time of session:** 60 min **Frequency:** 2x/week **Duration:** At least 12 weeks	**Primary:** well-being, QOL, body-image, and negative mood symptoms/maternal depression (WHOQOL-bref and SF-36). **Secondary:** pregnancy complaints, pelvic girdle pain, and LBP.	IG presented significantly lower values in the fatigue variable compared with CG. IG presented higher values in health satisfaction compared with CG. The IG presented lower values in variables of nausea/vomiting and numbness/reduced circulation compared with CG.
MK Gustafsson et al. (2015) [24]	**N =** 761 GI = 396 + GC = 365 Healthy PW without contraindications for practice (ACOG 2002). **Gestational Week:** 18th/22th **Location:** Trondheim, Norway.	To investigate whether a customized exercise program influences pregnant women’s psychological wellbeing and general health perception reflecting HRQoL in late pregnancy.	**Type of Intervention:** RCT (CG × IG) **Description:** Moderate intensity physical exercise program. Each session included aerobic and strength phases. The participants were also encouraged to exercise three times a week. The intervention was supervised by an exercise specialist. CG received standard antenatal care and the customary information, and they were not discouraged from exercise. Women in both groups received written standardized information and recommendations on diet, pelvic floor muscle exercises, and pregnancy-related pelvic girdle pain. **Intensity Measurement:** Borg Scale (12–14) Time of session: **Frequency:** 1x/week **Duration:** At least 12 weeks	**Primary:** self-perceived general health and psychological wellbeing before and after the intervention.	The study did not show significant differences in general health perception and psychological wellbeing in the third trimester between IG and CG.
Forouzan Charkamyani et al. (2019) [25]	**N =** 170 GI = 85 + GC = 85 Healthy PW IVF with no contraindications for practice (ACOG 2002). **Gestational Week:** 12th/16th **Location:** Tehran, Iran.	The role of a structured program of exercise training in low-risk pregnancy in Iranian women undergoing in vitro fertilization (IVF) based on the reduction of gestational diabetes was examined.	**Type of Intervention:** RCT (CG × IG) **Description:** Moderate intensity physical exercise program. Each session included walking, aerobic, strength and relaxation exercises. The intervention was supervised by an exercise specialist. A similar number of classes (1 weekly session for 90 days) for both the groups were held to present routine and general care in the pregnancy period relevant to the significant impacts of physical activity on maternal and fetal health. **Intensity Measurement:** Borg Scale (12–14) **Time of session:** 60 min **Frequency:** 3x/week **Duration:** 12 weeks	**Primary:** GDM, OGTT at the 24–28 and 34 gestation weeks. Gestational hypertension was measured in three distinct analyses after 20 week’s gestation.	The present study showed significant differences in the suitability of physical activity after and before intervention in IG. CG did not present differences. IG had a significant reduction in GDM. The physical exercise intervention can highly decrease the risk of developing pre-eclampsia.
María Perales et al. (2016) [26]	**N =** 241 GI = 120 + GC = 121 Healthy PW without contraindications for practice (ACOG 2002). **Gestational Week:** <16th **Location:** Madrid, Spain.	Investigate the effects of pregnancy exercise on echocardiographic indicators of hemodynamics, cardiac remodeling, left ventricular function, and cardiovascular disease risk factors.	**Type of Intervention:** RCT (CG × IG) **Description:** Moderate intensity physical exercise intervention. Each session included warm-up, aerobic, strength, and stretch exercises. The intervention was supervised by an exercise specialist. The duration of the aerobic and strength training component was kept constant. During the first trimester, more importance was given to improving body awareness, in the second trimester, the priority was to improve balance, in the third trimester, more emphasis was given to improving pelvis mobility. **Intensity Measurement:** Borg Scale **Time of session:** 55 to 60 min **Frequency:** 3x/week **Duration:** 18 weeks	**Secondary:** Baseline maternal characteristics, hypertension was measured at 20th and 34th gestational weeks. GDM was measured at 24th to 28th gestational weeks. GWG and depression levels were measured at the end of the intervention. Pregnancy outcomes were identified at the delivery.	The proportion of women with excessive GWG at the end of pregnancy in the IG compared with the CG was significantly lower. The values of the depression scale were lower in IG compared with CG.
Mireia Pelaez et al. (2019) [41]	**N =** 345 GI = 115 + GC = 230 Healthy PW without contraindications for practice (ACOG 2002). **Gestational Week:** 8th/10th **Location:** Arroyo, Spain.	To investigate the effect of supervised moderate to vigorous exercise on gestational weight gain, its related risks (gestational diabetes), macrosomia, and type of delivery), and the preventive effects on women who exceed the weight gain recommendations.	**Type of Intervention:** RCT (CG × IG) **Description:** Moderate intensity physical exercise intervention. Each session included warm-up, core exercises, major muscles groups resistance training, and relaxation phase. The intervention was supervised by an exercise specialist. Supine position, ballistic movements, and high-impact exercises were avoided. Group dynamics were used to enhance motivation and adherence (games, exercises in pairs or groups, social networks such as Facebook and WhatsApp). The CG received standard care and physical activity counseling from health care professionals. They were not discouraged from exercising on their own. **Intensity Measurement:** Borg Scale (12–14) **Time of session:** 60 to 65 min **Frequency:** 3x/week **Duration:** 24 weeks	**Primary:** GWG was measured at the first and the last prenatal visit. **Secondary:** GDM, macrosomia, and type of delivery.	The study showed that IG gained less weight than CG with a significant difference. IG was less likely to exceed the 2009 Institute of Medicine (IoM) recommendations than CG. The values of macrosomia were lower in IG than in CG. More normal vaginal deliveries were found in the IG than in the CG. In this study, the relationship between excessive GWG and the risks mentioned existed only in the CG and not in the IG, which leads us to hypothesize that exercise provides some level of protection against the risks associated with excessive GWG.
Erin Clark et al. (2018) [27]	**N =** 36 GI = 14 + GC = 22 Healthy PW without contraindications for practice (ACSM 2010). **Gestational Week:** ≤ 16th **Location:** United States of America.	To determine the influence of exercise on maternal lipid levels and infant body size.	**Type of Intervention:** RCT (CG × IG) **Description:** Moderate intensity physical exercise intervention. Each session included warm-up, aerobic, and relaxation phases. The intervention was supervised by an exercise specialist. The CG did not receive an exercise intervention period. **Intensity Measurement:** Borg Scale (12–14) **Time of session:** 60 min **Frequency:** 3x/week **Duration:** 22 weeks	**Primary:** GWG, Non-fasting lipid profiles, blood samples, serum samples, total cholesterol, HDL, and triglycerides were calculated at 16th and 36th weeks gestation. **Secondary:** Infant measures included gestational age at birth, delivery mode, sex, Apgar score at 1 and 5 min, birth weight, birth length, head, and abdominal circumferences.	The study showed that IG presented lower values of triglycerides compared with CG. The head circumference at birth has a positive relationship with exercise during pregnancy. Improved infant outcomes are associated with lower pre-pregnancy BMI, along with increased physical exercise levels during pregnancy and late pregnancy LDL levels. Birth weight and length are associated with the amount of aerobic exercise during pregnancy and maternal lipid levels. No differences in Apgar score between groups.
Lene A. H Haakstad et al. (2020) [28]	**N =** 105 GI = 52 + GC = 53 Healthy PW without contraindications for practice (ACOG 2002). **Gestational Week:** ≤12th **Location:** Olso, Norway.	To investigate the sole effect of supervised group exercise, including pelvic floor muscle training on the course of labor and mode of delivery.	**Type of Intervention:** RCT (CG × IG) **Description:** Moderate physical exercise intervention. Each session included warm-up, cardiovascular, strength, and relaxation phases. The intervention was supervised by an exercise specialist. **Intensity Measurement:** Borg Scale (12–14) **Time of session:** 60 min Frequency: **Duration:** At least 12 weeks. (36 sessions)	**Primary:** The course of labor and mode of delivery. **Secondary:** Baseline maternal characteristics.	In the mode of delivery, IG had more cesarean sections compared with the CG, without significant differences.
Raquel Rodríguez-Blanque et al. (2020) [29]	**N =** 129 GI = 65 + GC = 64 Healthy PW without contraindications for practice (ACOG 2015). **Gestational Week:** 20th **Location:** Granada, Spain.	To analyze the HRQoL in pregnancy for women who complete a program of moderate physical activity in the water.	**Type of Intervention:** RCT (CG × IG) **Description:** The intervention group had access to a moderate physical exercise program in water. The intervention was supervised by an exercise specialist. The CG received usual care and general advice on the benefits of exercise. Both groups received verbal and written dietary advice during pregnancy. **Intensity Measurement:** Borg Scale or Heart Rate **Time of session:** 60′ **Frequency:** 3x/week **Duration of Intervention:** 17 weeks	**Primary:** HRQol at the beginning and at the end of the intervention. **Secondary:** Sociodemographic and anthropometric variables were measured in the first and third trimesters and parity. Body weight was measured at weeks 12 and 36 of pregnancy. Perinatal results were identified after delivery. The level of physical activity was measured with a questionnaire at the beginning of the study.	The decrease in mean HRQol scores was significantly higher in the CG compared with IG. CG presented a higher risk of depression compared with IG.
Raquel Rodríguez-Blanque et al. (2020) [30]	**N =** 129 GI = 65 + GC = 64 Healthy PW without contraindications for practice (ACOG 2015). **Gestational Week:** 12th/20th **Location:** Granada, Spain.	To evaluate the prevalence of spontaneous birth among women who participated in a water-based physical exercise program.	**Type of Intervention:** RCT (CG × IG) **Description:** The intervention group had access to a moderate physical exercise program in water. The intervention was supervised by an exercise specialist. The CG received usual care and general advice on the benefits of exercise. Both groups received verbal and written dietary advice during pregnancy. **Intensity Measurement:** Borg Scale or Heart Rate **Time of session:** 60′ **Frequency:** 3x/week **Duration of Intervention:** 17 weeks	**Primary:** Intrapartum and neonatal outcomes (gestational age, reason for hospital admission, birth weight, and Apgar test). **Secondary:** sociodemographic and anthropometric variables were measured at the beginning and at the end of the intervention.	IG presented better control of GWG and higher rate of spontaneous birth and a lower rate of instrumental deliveries and cesarean sections. GWG is related to the Apgar score. PW with BMI is in the normal-weight range had more probably to have a baby with an Apgar score of 10 at five minutes. PW whose BMI is in the normal-weight range at the start of pregnancy are more likely to give birth spontaneously than those with overweight or obesity before pregnancy. An appropriate GWG is related to a physiological birth, while PW who present a higher GWG are more likely to require instrumental birth.
Olga Roldan-Reoyo et al. (2019) [31]	**N =** 131 GI = 64 + GC = 67 Healthy PW without contraindications for practice (ACOG 2002). **Gestational Week:** 10th/12th **Location:** Madrid, Spain.	To determine if regular maternal physical activity leads to measurable adaptations of the fetal autonomic nervous system measured by FHR response recovery time.	Intervention: RCT (CG × IG) **Description:** Moderate intensity physical exercise intervention. Each session included warm-up, aerobic, strength, pelvic floor, and stretching exercises. The intervention was supervised by an exercise specialist. **Intensity Measurement:** 40–60% heart rate reserve (HRR) **Time of session:** 60 min **Frequency:** 3x/week **Duration:** 28 weeks	**Primary:** FHR recovery time was measured between 34th and 36th gestational weeks (at 40% and 60% maternal HRR). **Secondary:** MHR recovery time, FHR at rest, FHR after exercise, and the difference between these timepoints.	This study showed that supervised moderate intensity exercise during pregnancy is associated with quicker FHR recovery time.
Raquel Rodríguez-Blanque et al. (2019) [32]	**N =** 129 GI = 65 + GC = 64 Healthy PW without contraindications for practice (ACOG 2015). **Gestational Week:** 12th **Location:** Granada, Spain.	To determine the effect of an aquatic physical exercise program performed during pregnancy on the rate of intact perineum after childbirth.	**Type of Intervention:** RCT (CG × IG) **Description:** The intervention group had access to moderate intensity physical exercise program in water. The intervention was supervised by an exercise specialist. The CG received usual care and general advice on the benefits of exercise. Both groups received verbal and written dietary advice during pregnancy. **Intensity Measurement:** Borg Scale or Heart Rate **Time of session:** 60′ **Frequency:** 3x/week **Duration of Intervention:** 17 weeks	**Primary:** Reason for admission, mode of labor, integrity of the perineum (intact, lacerations, episiotomy), gestation time, birth weight, and analgesia during labor. **Secondary:** Baseline maternal characteristics, GWG, level of physical activity (questionnaire).	The study showed that the IG presented lower values in birth weight compared with the CG, with a significant difference. In a variable of integrity of the perineum, the IG had significantly more incidences compared with the CG. The CG had more cases of lacerations and episiotomies but without significant differences.
M. Brik et al. (2019) [33]	**N =** 120 GI = 75 + GC = 45 Healthy PW without contraindications for practice (no guidelines). **Gestational Week:** 9th **Location:** Madrid, Spain.	To evaluate the association between physical exercise during pregnancy and maternal gestational weight gain and fetal cardiac function.	**Type of Intervention:** RCT (CG × IG) **Description:** Moderate intensity physical exercise intervention. Each session included warm-up, cardiovascular, strength, coordination and balance, pelvic floor and stretching exercises. The intervention was supervised by an exercise specialist. The CG were advised not to attend during pregnancy any supervised exercise program involving exercise for more than 30′ three times per week. However, they were not discouraged from exercising on their own. **Intensity Measurement:** 55 to 60% of Maximum Heart Rate **Time of session:** 60 min **Frequency:** 3x/week **Duration:** 29 weeks	**Primary:** GWG, fetal cardiac function parameters. **Secondary:** Baseline maternal characteristics and labor outcomes.	The physical exercise intervention did not control GWG but increased maternal weight loss after delivery. The physical exercise intervention did not affect fetal cardiac function.
Carolina de Vargas Nunes Coll et al. (2019) [12]	**N =** 639 GI = 75 + GC = 45 Healthy PW without contraindications for practice (ACOG 2015). **Gestational Week:** 16th/20th **Location:** Rio Grande, Brazil.	To assess the efficacy of regular exercise during pregnancy in the prevention of postpartum depression.	**Type of Intervention:** RCT (CG × IG) **Description:** Moderate intensity physical exercise intervention. Each session included warm-up, aerobic, strength and relaxation exercises. The intensity was managed with the evolution of pregnancy. The intervention was supervised by an exercise specialist. The CG was advised to maintain their usual daily activities. **Intensity Measurement:** Borg Scale (12–14) **Time of session:** 60 min **Frequency:** 3x/week **Duration:** At least 16 weeks	**Primary:** Self-reported postpartum depressive symptoms (questionnaire). **Secondary:** Baseline maternal characteristics.	The study did not show significant differences between groups in postpartum depression. CG presented a higher risk of depression compared with IG.
Raquel Rodríguez-Blanque et al. (2019) [34]	**N =** 140 GI = 70 + GC = 70 Healthy PW without contraindications for practice (ACOG 2015). **Gestational Week:** 20th **Location:** Granada, Spain.	To determine the duration of labor in pregnant women who completed a program of moderate physical exercise in water and subsequently presented eutocic birth.	**Type of Intervention:** RCT (CG × IG) **Description:** The intervention group had access to moderate physical exercise program in water. The intervention was supervised by an exercise specialist. The CG received usual care and general advice on the benefits of exercise. Both groups received verbal and written dietary advice during pregnancy. **Intensity Measurement:** Borg Scale or Heart Rate **Time of session:** 60′ **Frequency:** 3x/week **Duration of Intervention:** 17 weeks	**Primary:** birthweight and duration of 1st, 2nd, and 3rd stages of labor, and the type of labor. **Secondary:** Baseline maternal characteristics.	The study showed that neonatal birth weight was significantly lower in IG than in CG. The study presented that the 1^st^ and 2^nd^ stages of labor were shorter for PW who performed intervention. The total delivery time for IG was almost 3 h less than in CG.
Eman Awad et al. (2019) [35]	**N =** 60 GI = 30 + GC = 30 Healthy PW without contraindications for practice (ACOG 2002). **Gestational Week:** 24th **Location:** Cairo, Egypt.	To determine the effect of an exercise program on the mode of delivery in gestational diabetic females	**Type of Intervention:** RCT (CG × IG) **Description:** Moderate intensity physical exercise intervention. Each session included warm-up, aerobic, strength and relaxation exercises. Both groups received the same diet with insulin therapy. Intensity Measurement: **Time of session:** 60 min **Frequency:** 3x /week **Duration:** 12 weeks	**Primary:** Mode of delivery and Apgar score. **Secondary:** Baseline maternal characteristics.	The study showed a significant decrease in the number of cesarean deliveries in the IG compared with the CG. The neonates of IG had Apgar score at 1^st^ and 5th minutes after delivery better compared with CG.
Ruben Barakat et al. (2017) [5]	**N =** 65 GI = 33 + GC = 32 Healthy PW without contraindications for practice (ACOG 2015). **Gestational Week:** 8th/11th **Location:** Madrid, Spain.	To examine the influence of an aerobic exercise program throughout pregnancy on PW among healthy pregnant women.	**Type of Intervention:** RCT (CG × IG) **Description:** Moderate intensity physical exercise intervention. Each session included warm-up, aerobic dance, strength exercises, pelvic floor muscle training, and relaxation exercises. All sessions were supervised by a qualified fitness specialist and with an obstetrician’s assistance. The CG received general advice about the positive effects of physical activity. **Intensity Measurement:** Borg Scale (12–14) and Heart Rate (55–60% of HRR) **Time of session:** 55 to 60 min **Frequency:** 3x/week **Duration:** 28 weeks. (84 sessions)	**Primary:** The placental weight was measured during the first 30′ after delivery. **Secondary:** Gestational age, type of delivery, body weight, Apgar score, GDM, and hypertension.	The study did not present differences between both groups in variables. The study showed that supervised moderate exercise training during pregnancy did not affect negatively placental weight, the overall health status of the newborn, and Apgar score.
Now it´s corAlka Pawalia et al. (2017) [36]	**N =** 36 GI = 12 + GC = 12 Healthy PW without contraindications for practice (ACOG 2015). **Gestational Week:** 16th **Location:** Haryana, India.	To investigate the effect of physical activity and diet during the prenatal period and its effect on gestational weight gain (GWG), BMI, waist circumference (WC), hip circumference (HC), and postpartum weight retention (PPWR).	**Type of Intervention:** RCT (CG × IG x IDG) **Description:** Moderate intensity physical exercise intervention. Each session included warm-up, pelvic floor and Kegel’s exercises, abdominal and back care exercises, and relaxation and meditation exercises. The IG was asked to do the same exercises at home for at least 3 day/week apart from the supervised session and were also encouraged to walk daily for a minimum duration of 30 min and to do so at least 4 days/week throughout pregnancy. The IDG received timely telephonic messages emphasizing the need for adequate and healthy food choices to be followed during pregnancy. The CG was advised once at recruitment for following proper diet care and explained the importance of being physically active during pregnancy. The intervention was extended to 2 months after delivery. **Intensity Measurement:** Borg Scale (12–14) **Time of session:** 60 to 90 min **Frequency:** 2x/week **Duration:** 30 weeks	**Primary:** GWG, BW, fetal waist circumference, hip circumference, waist to hip ratio, and maternal weight retention at postpartum. **Secondary:** Baseline maternal characteristics.	The study showed birth weight was significantly lower in IG and IDG compared with CG. The IG presented lower values on the waist circumference, which has a direct relation with lifestyle diseases. There were no significant differences in GWG and Maternal weight retention at postpartum between groups.
Ruben Barakat et al. (2016) [4]	**N =** 840 GI = 420 + GC = 420 Healthy PW without contraindications for practice (ACOG 2002). **Gestational Week:** 9th/11th **Location:** Madrid, Spain.	To examine the impact of a program of supervised exercise throughout pregnancy on the incidence of pregnancy-induced hypertension.	**Type of Intervention:** RCT (CG × IG) **Description:** Moderate intensity physical exercise intervention. Each session included warm-up, strength exercises, and a relaxation phase. The CG received general advice about the positive effects of physical activity. The CG was asked by telephone about their exercise during pregnancy with a questionnaire. If they were active during pregnancy, they were excluded. **Intensity Measurement:** Borg Scale (12–14) and Heart rate (70% of MHR) **Time of session:** 50 to 55 min **Frequency:** 3x/week **Duration:** 28 weeks	**Primary:** Number of women who developed hypertension during pregnancy, diastolic and systolic arterial blood pressure (every visit), and BW. **Secondary:** Baseline maternal characteristics and GWG.	Significantly lower values on the incidence of hypertension in IG. Excessive GWG, gestational diabetes, and preeclampsia was significantly prevented in IG. Physical exercise intervention also reduced the incidence of macrosomia and protected against low-birthweight infants.
Yaiza Cordero et al. (2015) [42]	**N =** 257 GI = 101 + GC = 156 Healthy PW without contraindications for practice (no guidelines). **Gestational Week:** 10th/12th **Location:** Madrid, Spain.	To assess the effectiveness of a maternal exercise program (land/aquatic activities, both aerobic and muscular conditioning) in preventing gestational diabetes mellitus (GDM).	**Type of Intervention:** RCT (CG × IG) **Description:** Moderate intensity physical exercise intervention in and out of water. Each land session (2x) included activation, physical and psychological preparation, aerobic choreography, strength exercise, pelvic floor muscle training, and stretching. Each water session (1x) included warm-up, core and strength exercises and stretching exercises. Water Temperature: 28.5–29° **Intensity Measurement:** Borg Scale (12–14) **Time of session:** 50 to 60 min **Frequency:** 3x/week **Duration:** 10–12 weeks to the end of the pregnancy.	**Primary:** GDM **Secondary:** Baseline maternal characteristics, GWG, gestational age at delivery, type of delivery, BW, and length.	The study showed that physical exercise intervention was strongly associated with a decrease in GWG, and preserved glucose tolerance.

BMI—body mass index; CG—control group; FHR—fetal heart rate; GDM—gestational diabetes mellitus; GPAQ—global physical activity questionnaire; GWG—gestational weight gain; HC—hip circumference; HDL—high-density lipoprotein; HRQoL—health-related quality of life; HRR—heart rate reserve; IDG—intervention and dietary group; IG—intervention group; LBP—low back pain; LDL—low-density lipoprotein; MHR—maternal heart rate; OGTT—oral glucose tolerance test; PFM—pelvic floor muscles; PPD—postpartum depression; PW—pregnant women; PPWR—postpartum weight retention; WC—waist circumference.

### 3.3. Characteristics of the Interventions

All studies aimed to evaluate the effectiveness of a group exercise program on maternal and fetal health outcomes. The interventions had between 24 to 90 group sessions of an average of 8–13 women. The sessions had a frequency of at least one to four times per week and an average duration of 60 min. Only one study presented sessions with a duration of between 60 and 90 min. The intensity of the intervention was measured with Borg’s scale remaining between 12 and 14, or with heart rate remaining at maximum values of 60% [5,31,33] or 70% [4]. One study used heart rate with women who reported a value higher than 14 on the Borg’s Scale [29].

The structure of interventions developed out of water consisted of a warm-up period with low-impact aerobic exercises such as walking and dancing, the main phase with resistance training, pelvic floor muscle training, and ended with relaxation, stretching, flexibility, and mobility. Nine studies included an aquatic intervention [17,20,29,30,32,34,38,39,42]. Specifically, this method consisted of three sessions of 60 min per week. Each session had one day of recovery. Nutrition and hydration were essential before sessions. The temperature of the water was between 28–30 degrees Celsius. Each session was divided into three phases: the warm-up phase (in and out of water); the aim phase, first in 1.8 m (or 5 ft) and after in 1.2 m (or 3 ft) m pools, that consisted of resistance training using the four swimming techniques and pelvic floor muscle training, respectively; finally, the women had a relaxation phase. The intensity is measured with Borg’s scale, heart rate with a maximum of 140 bpm, or the talk-test. The level of training was measured with IPAQ—an international physical activity questionnaire, using an accelerometer for seven days, and by the mastery of the swimming techniques.

Only one study used both methods and women had two sessions in the gym hall and one session in the pool. In the gym, the intervention was structured like in the previously mentioned studies. In the pool, the intervention was focused on each of the swimming techniques (excluding the butterfly technique). The pelvic floor muscle training was mentioned only in the gym hall sessions [42].

In the studies that used a gym-based method, the intervention group received an initial telephone consultation with a doctor, clinical nutritionist, or graduate student in public health. A follow-up consultation took place 4–6 weeks after the initial consultation. They had other nutrition initiatives during pregnancy.

The physical exercise interventions were supervised and led by certified aerobic/fitness instructors or by physiotherapists. The control groups included pregnant women receiving standard antenatal care who were neither encouraged nor discouraged from exercising. Some studies reported that the control group had additional written information and recommendations about diet, pelvic floor muscle exercises, and pregnancy-related pelvic girdle pain [7,18,22,37].

#### CERT Model

On analysis of the intervention’s design with the CERT Model, a total score was assigned by summing all responses, as presented in Table 3. Briefly, 9 articles had a 12 or more total score. The maximum score was 15 and the minimum score was 5. Only one study recorded the minimum score.

### 3.4. Maternal and Fetal Physical Activity and Health Outcomes

The following effects of the exercise intervention on maternal and fetal health and fitness outcomes were analyzed:


**Maternal Outcomes (Primary)**
Gestational Weight Gain (GWG) [9,18,19,20,26,27,30,33,36,39,41]Fat percentage at 36 gestational weeks [19,27]Non-fasting lipid profiles, blood samples, serum samples, total cholesterol, HDL, triglycerides [27]Gestational diabetes and/or 1 h Oral Glucose Tolerance Test [25,40,42]Gestational Hypertension and Pre-eclampsia [8]Resting and Walking systolic and diastolic Blood pressure [4]Maternal heart rate recovery [31]Pelvic Floor Muscle strength [22]Perineal status after birth [32]Quality of life [23,24,29]Sleep quality [38]Incidence of operative deliveries [18]Type and duration of labor [28,30,32,34,35,37]Weight retention at postpartum [7,36]Prevalence of Postpartum Depression (PPD), Depression [12,20,21,23,24,26,29]



**Fetal Outcomes (Primary)**
Head Circumference [27]Newborn weight [7,9,17,30,32,34,39]Neonate with Macrosomia [40,41]Apgar Score [30,35]Fetal Heart Rate recovery time [31].


### 3.5. Effectiveness of the Interventions

The studies reported the following main and significant effects on maternal outcomes of the interventions:
The control group reported higher gestational weight gain [4,18,19,20,26,30,39,40,41,42,43]; and higher values of triglycerides at 16 and 36 weeks [27]Faster maternal heart rate recovery in the intervention group [31]The intervention group presented lower weight retention at four and seven months of postpartum [4]Physical activity during pregnancy increases weight loss after delivery [33]The ratio of women diagnosed with gestational diabetes mellitus was higher in the control group [4,25,40,42]; and the results of 1 h oral glucose tolerance test were lower in the intervention group [40]Blood pressure at rest and during exercise were lower in the intervention group [7]; and the incidence of preeclampsia and hypertension was higher in the control group [4,25]Strength, endurance, and repeatability of pelvic floor muscle were higher in the intervention group [22]; furthermore, the intervention group presented fewer lacerations and episiotomies [32]Physical activity intervention decreased the levels of depression during pregnancy [26,29,30], and is linked with a lower incidence of PPD [14,21]; with significant differences in the overweight and obesity categories [20]Higher rate of spontaneous birth [28,30,41]; and a lower rate of instrumental deliveries and cesarean sections in the intervention group [30]The intervention group presented shorter first and second stages of labor [34,37]; and physical activity was associated with lower odds of acute cesarean delivery [30,35,40]Levels of fatigue, nausea/vomiting, and numbness/reduced circulation were higher in the control group [23]; additionally, the control group reported worse values for quality, duration, latency, and regular efficiency of sleep [38]Decreased quality of life was higher in the control group [29].

Significant effects on newborn outcomes of the interventions were reported by the studies too: Newborn weight was lower in the intervention group [4,32,34,36,43]. However, one study reported that despite the differences, both groups were in a normal state of weight [17]; The ratio of neonate macrosomia was higher in the control group [5,40,41]The head circumference at birth has a positive relationship with exercise during pregnancy [27]Better results of Apgar Score were associated with women who practiced physical activity or that had normal weight during pregnancy [30,35]The fetal heart rate recovery time is associated with moderate-intensity exercise during pregnancy [31].

## 4. Discussion

Group exercise is quite popular and promotes socialization [21] and adherence [13] among pregnant women. The present systematic review aims to update the knowledge about the effectiveness of group exercise during pregnancy in improving women’s health outcomes during pregnancy, as well as a newborn’s health outcomes. Additionally, the authors analyzed the physical exercise program design and the consistency with the guidelines, identifying limitations in the interventions that can be improved in the future. In our previous review [15], a total of 17 studies were selected for analysis. All studies were RCT conducted with pregnant women that evaluated the effect of group exercise programs on the health outcomes of mother and newborn. These studies were conducted between 2009 and 2014, based on the ACOG guidelines published in 2002 [44]. New guidelines were published in 2015, affecting clinical, research, and exercise practices regarding physical activity during pregnancy. These are the reasons for not including trials conducted prior to 2015.

Thirty-one group exercise-based RCTs published in 2015, were selected for analysis in this review, to understand the effectiveness of group exercise programs in improving women’s health outcomes during pregnancy, as well as newborns’ health outcomes.

Considering the aims of this systematic review, the main outcome variables in the analysis are related to maternal health (i.e., quality of life, weight gain, gestational diabetes, cholesterol, hypertension, pre-eclampsia, well-being, depression, rest heart rate, sleep quality, low back pain), maternal physical activity (i.e., level and type of physical activity), and maternal fitness (i.e., cardiorespiratory fitness, strength, flexibility, balance, coordination, pelvic floor muscle strength), and labor outcomes (i.e., mode of delivery, birth weight, fetal cardiac function, and Apgar score).

Regarding the 31 RCT studies analyzed in the present review, the authors investigated if the studies followed the ACOG [2,44,45] or the Canadian guidelines [1], having in mind the need for consistency between practical interventions. Six studies did not present the updated guidelines [25,28,31,35,39,41], and three studies do not even refer to the guidelines as the basis of their interventions [27,33,42].

Several systematic reviews support the promotion of moderate-to-vigorous prenatal physical activity for maternal health benefits [3,15,46,47,48,49,50,51,52,53,54,55]. Indeed, the maternal well-being and quality of life are improved when women participate in regular group exercise during pregnancy, as shown by large RCTs led by Stafne et al. (2012) [56], Ruiz et al. (2013) [57], Haakstad et al. (2015) [23], Sagedal et al. (2016) [58], and Barakat et al. (2016; 2019) [4,40].

In general, the effectiveness of these studies is in line with the literature, with physical exercise contributing to the improvement of maternal and fetal health. Regarding maternal outcomes, the intervention groups reported more control of maternal weight gain, lower risk of GDM, lower risk of preeclampsia and hypertension, lower blood pressure in rest and during exercise, lower levels of depression, fewer cesarean deliveries, shorter first and second stages of labor, and better quality of life. On fetal outcomes, the intervention groups presented better control of newborn weight gain and a lower ratio of neonate macrosomia, better results on the Apgar score, and faster recovery of fetal heart rate. Thus, the findings of this systematic review support group exercise for the above-mentioned outcomes. Nevertheless, a gap in the literature was found in recent years regarding various maternal physical fitness parameters (i.e., cardiorespiratory fitness, strength, flexibility, balance, coordination).

In accordance with the guidelines [45], all the studies reported that certified fitness instructors supervised the exercise program, and some studies presented the qualifications of the physical exercise specialists. Most of the exercise programs were structured with: a warm-up (low impact aerobic exercise); the main phase (aerobic and resistance training); a pelvic floor muscle training phase; and, a stretching and relaxation phase. Although the literature suggests that combining aerobic exercise and resistance training during pregnancy is more effective at improving health outcomes than interventions focused on aerobic exercise alone, one study had only aerobic exercise [27].

Most studies compared the variables between the intervention group that received a supervised exercise program and the control group that received usual care. Only one study used three groups to evaluate the nutrition variable (Intervention and Dietary Group) [36].

The results of most of the studies were in line with the literature, with physical exercise contributing to the improvement of maternal and fetal health. Regarding maternal outcomes, the intervention groups reported more control of maternal weight gain, lower risk of GDM, lower risk of preeclampsia and hypertension, lower blood pressure in rest and during exercise, lower levels of depression, fewer cesarean deliveries, shorter first and second stages of labor and better quality of life. On fetal outcomes, the intervention groups presented better control of newborn weight gain and a lower ratio of neonate macrosomia, better results on the Apgar score, and faster recovery fetal heart rate. Maternal weight was often associated with other complications in pregnancy.

Moreover, a recent review identified important gaps in research, including a lack of studies investigating the benefits of group interventions [59]. As far as we are concerned, there are no studies investigating the benefits of group interventions with pregnant women when compared with other types. The novelty of this review is the inclusion of group exercise programs, and the use of CERT to assess those programs.

The results of the main variables will be presented and discussed, as well as possible failures in the analysis of some of these or limitations in the literature, as follows.

### 4.1. Maternal Weight Gain

Gestational weight gain can occur due to maternal fat accumulation, fluid expansion, and fetus, placenta, and uterus growth. Gestational weight gain is necessary for a fetus’ health, but it should be controlled, not reporting other adverse outcomes [60]. Excessive weight gain increases the risk of GDM, hypertension, and preeclampsia [61]. So, maternal weight can be a great indicator of maternal health in pregnancy. Most of the studies in this systematic review reported the significant impact of an exercise intervention on the control of weight gain [4,18,19,26,30,40,41,42,43]. Moreover, it was the variable that presented the most significant results in the studies included in this systematic review.

### 4.2. Gestational Diabetes

It is known that the placenta produces hormones responsible for the growth and development of the newborn. However, these hormones may promote insulin resistance in the mother’s cells, and it can consequently influence the GDM emergence. GDM is associated with adverse outcomes for the mother and the newborn [35]. Diet and exercise are used to prevent or manage it once it occurs [61]. In this review, the studies focused on the prevention of GDM, and just manipulate the physical exercise variable. The four studies that evaluated the impact of physical exercise programs reported a lower risk of GDM in the Intervention Group compared with the Control Group with statistically significant differences [4,25,40,42].

Cordero et al. reflected at the end of the RCT on the possible importance of the relationship between the timing of the beginning of the intervention and the efficacy of GDM prevention, reporting that the earlier the intervention starts, the lower the risk of GDM [42]. Every study of the present RCT began the intervention during the first trimester. In this trimester, the levels of the hormones change and increase dramatically [61]. So, in this period, the control of insulin is relevant. It can be relevant to understand the influence of this timing in the future.

Furthermore, it is known that a history of glucose intolerance, past gestational diabetes, or having first-degree relatives with diabetes are also associated with a higher risk of developing GDM. Some studies, specifically those that study the GDM variable, did not consider some of these factors that can influence the results [4,25,40,42].

It is important to refer, too, that the nutrition variable can be relevant to GDM results. Some studies reported the lack of nutritional monitoring as a limitation [5,40]. Spaight et al. exposed some studies that evaluated the association between nutrition in pre-pregnancy and the risk of GDM. Interestingly, the findings suggested that animal protein intake is associated with a lower risk of GDM and inversely associated with vegetal protein. The diets with low fiber and high glycemic load and high intake of animal fat and cholesterol were associated too. If the pre-pregnancy habits can influence the risk of GDM in the future, controlling this variable through physical exercise intervention will be necessary [62].

The ACOG warned of the risk of hypoglycemia on high-intensity or prolonged exercise of more than 45 min [45]. Pregnant women should be advised to adequate caloric intake before exercise or limit the intensity to minimize this risk [1].

### 4.3. Pre-eclampsia and Hypertension

The relationship between hypertensive disorders and physical exercise is clear [48,49]. In 2017, Gregg and Ferguson presented an unclear view with some reviews that reported any significant impact of moderate-intensity physical exercise during pregnancy on the risk of hypertension, and other reviews that inversely presented a positive impact [63]. In the present review, two studies reported a significant reduction in the risk of hypertension [4,25]. One study presented lower values of blood pressure in the Intervention Group compared to the Control Group, between, before, and after the intervention, although without significant differences [6]. The ACOG reported a reduction of hypertensive disorders as a benefit of physical exercise, despite presenting some studies without significant results in this variable [45]. Considering this systematic review, more studies are needed to clarify the relationship between exercise and hypertension.

### 4.4. Depression

The PPD [20,21] and/or depression [21,26] variables were evaluated in four studies. Most studies reported a positive association between physical exercise and the prevention of PPD and depression. As said before, maternal weight can trigger other risks for pregnancy. Aguilar-Cordero et al. evaluated the prevalence of PPD with an EPDS questionnaire. The study concluded that all obese or overweight women in the third trimester are at higher risk of PPD, but this risk is lower among those who follow a physical exercise program [20].

Although the results reported were positive, the evaluation of this variable can be biased. The sessions were in a group setting, and this factor could have a preventive impact on depressive symptomatology, whereas these classes promote socialization [21]. The difference in the impact of social factors and physical exercise on the risk of depression should be clarified. In the future, it is convenient that the analysis of the impact of the socialization variable in physical exercise programs on the prevention of PPD.

### 4.5. Type and Duration of Delivery

Prenatal exercise reduced the odds of instrumental delivery in the general obstetrical population [64]. In the present review, the type of labor was evaluated, and the studies reported a higher rate of spontaneous birth and a lower rate of instrumental deliveries and cesarean sections in the intervention group [28,30,35,37,41]. The studies that evaluated the duration of labor revealed that the intervention group had shorter first and second stages of labor [34,37].

### 4.6. Quality of Life

The World Health Organization (WHO) had a broad and multidimensional definition of quality of life that incorporates physical, psychological, social, and environmental aspects of life. In this systematic review, the three studies that evaluated this variable used different questionnaires that enable the assessment of the impact of mental and physical health status on different areas in a person’s life: General Health Questionnaire (GHQ) [64], SF-36 [65] and Psychological General Well-being Index (PGWBI) [66]. Two studies reported better results in the Intervention Group compared with the Control Group. Gustafsson et al. [24] did not report differences between groups. It is important to note that this intervention was performed with a considerably lower training frequency compared to the other studies in the systematic review (1x/week). Comparing the variable evaluation method between studies, in addition to the discrepancy between the questionnaires used, there is no questionnaire measuring this outcome specifically in pregnancy. In the future, it is relevant to adapt these questionnaires to this type of population [23,24,29].

### 4.7. Newborn Weight Gain and Macrosomia

Davenport et al. [67] showed that prenatal exercise is safe and beneficial for the fetus. Maternal exercise was associated with reduced odds of macrosomia (abnormally large babies) and was not associated with neonatal complications or adverse childhood outcomes The present systematic review presented five studies with statistically significant differences between both groups on the birth weight variable [4,17,32,34,36,43]. The moderate physical exercise program positively impacted birth weight and any adverse effects on fetal health. Furthermore, one study reported higher significant values of macrosomia in the control group [5].

### 4.8. Apgar Score

The Apgar score provides an accepted and convenient method for reporting the status of the newborn infant immediately after birth and adaptation to extrauterine life [68]. The variables were measured at 1 min and 5 min after delivery. The test score ranges from 0–10. The normal maternal weight gain was associated with a score of 10 at 5 min [30]. The intervention group presented a significantly better score at both times, but no groups presented low values on the Apgar Test [35]. Moderate-intensity physical exercise is associated with a better Apgar score, although the research does not associate a low score (<7) to sedentarism in pregnancy.

### 4.9. Fetal Heart Rate Recovery

Roldan-Reoyo et al. [31] evaluated the Fetal Heart Rate recovery variable. The participants performed two walking tests (40% and 60% of Maternal Heart Rate Reserve), and it was measured at the beginning and after 20 min of the maternal walking test. On both intensities, the fetuses of the Intervention Group were quicker to the recovery heart rate. Additionally, after the 60% intensity walking test, the difference in fetuses that recovered within 20 min between both groups was statistically significant (100% of fetuses in the Intervention Group recovered within the time). Although the literature is scarce, it presents a positive relationship between exercise and fetal heart rate recovery.

### 4.10. Pelvic Floor Muscles Training (PFM)

In pregnancy, due to hormonal changes, it is possible to have a PFM strength reduction that may result in pelvic floor dysfunction. Preventive strategies such as PFM training during pregnancy can be the solution for possible disorders (the most common is urinary incontinence) [22]. The Canadian Guidelines reported the importance of pelvic floor muscle training to reduce the risk of urinary incontinence, although still supported by weak recommendations and low-quality evidence [1].

Pires evaluated the effectiveness of PFM supervised training and another PFM unsupervised at home. The intervention began in the 28th gestational week and used digital palpation, the Oxford Grading Scale to evaluate the muscle contractility, and the Pad test to quantify the urine loss. This study did not use the manometer evaluation. The study reported significant improvements in pregnant women [69]. A systematic review evaluated the effectiveness of PFM training on the risk of urinary incontinence, and the results reported that PFM training decreased the risk of urinary incontinence [70].

In this systematic review, only one study evaluated the pelvic floor muscle strength through manometer and digital palpation. The physical exercise intervention finished on the 32–34th gestational week. The results did not report differences in the strength of pelvic floor muscles on manometer evaluation, but it presented improvement of digital palpation variables (PFM strength, PFM endurance, and PFM repeatability) with significant differences between intervention and control groups [22].

It is already known that the Pilates method improves control over the core muscles, strengthening them. Due to the great popularity of this method throughout the world over the last decade, it may be relevant for more literature that corroborates the benefits of pelvic strength work associated with it.

### 4.11. Level of Physical Activity

The ACOG affirmed that pregnancy is an ideal time for behavior modification and adoption of a healthy lifestyle [2]. Evaluating the physical activity level (excluding the exercise program) before, during, and after pregnancy can be a way to evaluate the impact of a physical exercise intervention on lifestyle habits. All studies that reported this evaluation presented increasing physical activity levels in the exercise group and inversely in the control group [25]. Some studies counseled the participants to do some physical exercise (e.g., 3 days/week of moderate-intensity physical exercise) at home to complement the supervised program, but this variable was not considered in the analysis [19]. Due to the lack of consensus in studying this variable, the effectiveness of the home program/physical activity combined with supervised physical exercise and its influence on the results is unclear. It is important to develop other intervention studies that analyze this variable. 

### 4.12. Adherence of Program

The program’s adherence revealed itself as a limitation and a significant challenge in many RCTs [13]. Some studies reported that this is due to poor follow-up of the control group compared to the intervention group. Other studies presented some adverse events of pregnancy as reasons for dropout. One group reported high adherence to the program justifying the supervision of the programs carried out by qualified exercise professionals ensuring safety to the participants. If the adverse events and poor follow-up are reasons for the dropout of participants, it is important that interventions had a multidisciplinary team to accompany the participants to guarantee safety and motivation. Barakat et al. reported that an obstetrician accompanied the physical exercise intervention. This study had >85% adherence, justified by the authors by the variety of exercises. However, it is important to note that there was no dropout due to pregnancy-related complications, probably due to the presence of a multidisciplinary team [5]. Rodríguez-Blanque et al. [29] reported that the difficulty in recruiting sufficient participants was due to, in part, the health services involved did not provide sufficient, appropriate information to respond to doubts raised by some pregnant women regarding physical exercise during pregnancy. This problem reminds us of the need for a common language between the various stakeholders. Furthermore, the ACOG reported that if the pregnant woman had an absolute or relative contraindication for exercise, she should continue to perform, at least, some resistance training [2]. The lack of compliance may be related to the lack of a multidisciplinary team.

### 4.13. Ratings of Perceived Exertion vs. Heart Rate

Due to blunted and normal heart rate responses to exercise reported in pregnant women, the guidelines suggested that ratings of perceived exertion may be a more effective means to monitor exercise intensity than heart rate parameters [10]. Nonetheless, some studies have used the heart rate method [4,5,17,20,21,30,31,32,33,38,43]. This adaptation should be further studied and clarified in future research.

### 4.14. CERT Evaluation

The CERT model evaluated the report of the exercise intervention design. The main limitations found in the evaluated interventions, which can compromise their clarity and replication, are reported as follows:Few studies present motivation strategies. Conversely, they present a high dropout of participantsFew studies report a final reflection of the intervention and the fulfillment of the initial planningFew studies have controlled the physical activity performed beyond the exercise intervention. Two problems can arise from this limitation: difficulty in evaluating the effectiveness of a program performed at home and the impossibility of evaluating its influence on the progress of pregnancyFew studies present the program in detail, reporting intensity monitoring, frequency, duration, and detailed description of the type of exercises, making it difficult to replicate the exercise program in future interventionsFew studies assess the initial level of participants to adapt the intervention to each pregnant womanFew studies clarified the monitoring and evaluation of other components than exercise that can influence the results, such as nutrition.

## 5. Future Research

Due to the limitations mentioned above, a few proposals for future studies were discussed as follows.

Cai et al. [71] concluded that prenatal exercise interventions improve maternal predicted/measured cardiorespiratory fitness in pregnancy. However, this meta-analysis highlighted the need for additional high-quality studies. There is a lack of studies examining the effectiveness of group exercise programs on the levels of maternal physical fitness parameters, such as muscular fitness, balance, and coordination. Thus, those parameters should be addressed in future interventions.

The Canadian Guidelines state that, as well as the pregnant woman’s weight, diabetes, and age, a pregnant woman’s level of physical activity before pregnancy can also be a risk factor. In the future, studies that assess the influence of the level of physical activity and the practice of physical exercise during pre-pregnancy on the quality of life of pregnant women will be convenient.

The literature argues that pregnancy can be a time for adopting healthy practices. In the future, it would be of serious interest to analyze if previously inactive pregnant women who participate in intervention programs maintain exercise levels after pregnancy.

The social factor can also be a determinant of depression levels during and after pregnancy. In this systematic review, most studies showed a positive influence between the exercise program and depression levels. However, all studies included group interventions, which can bias the results. It would be interesting to compare group and individualized interventions in future research, as well as in-person versus virtual interventions. Trials comparing the effectiveness of in-person with online interventions, that were substantially increased after the COVID-19 pandemic in 2020 [72,73], would be of particular interest.

As mentioned before, the ACOG states that the relative and absolute contraindications may not justify stopping the practice of exercise, namely the strength training and pelvic floor. No study presented an alternative intervention plan for these hypotheses to promote the continuation of physical exercise in women with pregnancy-related complications. In the future, it would be convenient for a multidisciplinary follow-up to supervise these groups and ensure more safety and confidence (e.g., gestational diabetes, depression, among other medical conditions), promoting a high adherence.

The authors suggest creating or adapting existing questionnaires that assess the quality of life and well-being during pregnancy. This validation will, consequently, bring more accuracy to the results presented.

Chan et al. [74] reported that physical activity and exercise interventions lack standardized methodologies in the development, delivery, and assessment of such programs due to their complexity, regarding the different interacting components, interprofessional, and intervention contexts. The non-compliance with the CERT model in all articles is a limitation for the replication and accuracy of the exercise programs. In the future, the authors consider it relevant that exercise programs comply with the CERT model. Moreover, it would be interesting to develop and evaluate prenatal exercise interventions included in trials, as complex interventions, as proposed by Möhler et al. [75] and Skivington et al. [76].

The main strengths of this systematic review are that it included only RCT studies with a high level of quality. The CERT model that assessed the intervention program allowed us to perceive some gaps in the interventions. The main limitations are that even using important databases, there is a possibility that other relevant studies were not included.

## 6. Conclusions

The present systematic review reported significant effects of the group exercise programs on maternal health status, specifically, lower maternal weight gain, faster heart rate recovery, improved glucose tolerance, lower blood pressure, better performance of pelvic floor muscles, decreased risk of depression, higher rate of spontaneous birth, lower complications during labor, lower newborn weight, less neonate macrosomia, faster fetal heart rate recovery time and better results on the Apgar score. These results are in line with the results of the original systematic review, and support the positive relationship between physical exercise and pregnancy, helping to dispel some myths. Nevertheless, there should be more consistency between the variables of physical exercise used in the different interventions, according to the current guidelines. Some suggestions were presented for future studies or improvements regarding prenatal exercise interventions. Exercise and health professionals should inform pregnant women about the benefits for maternal and fetal health and should advise pregnant women regarding the practice of physical exercise in groups during pregnancy, in compliance with the guidelines. Preferably, that practice should be accompanied by a multidisciplinary team to convey more confidence to the pregnant woman.

## Figures and Tables

**Figure 1 ijerph-19-04875-f001:**
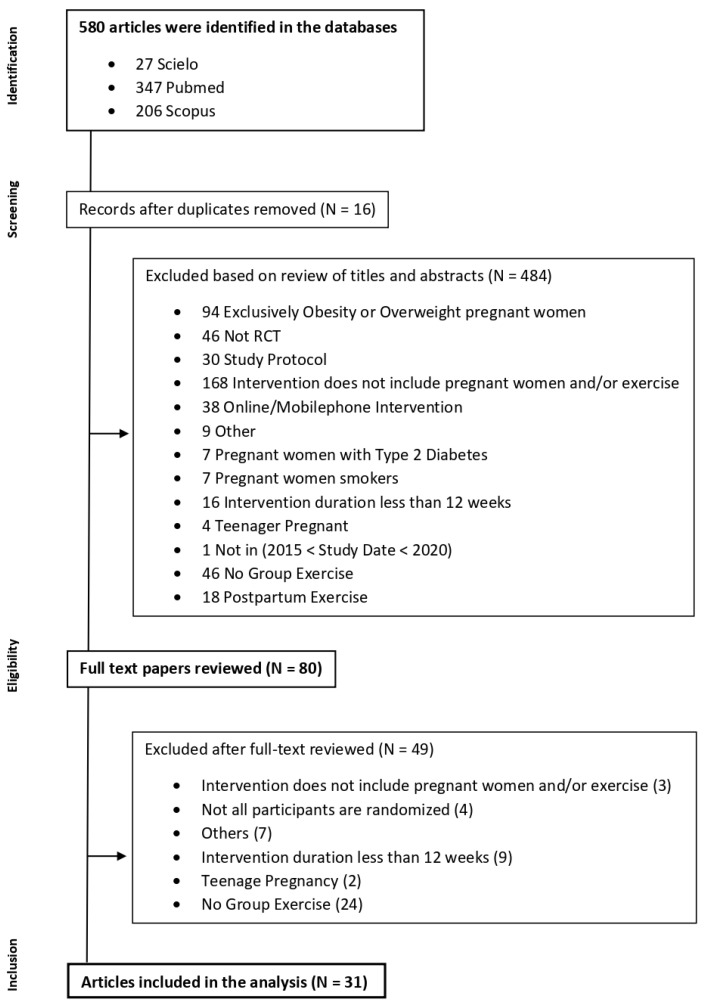
Flow diagram of literature search for group exercise interventions in pregnancy.

**Table 3 ijerph-19-04875-t003:** CERT Model.

Authors	Total	Authors	Total	Authors	Total
Sánchez-García, J. (2019) [43]	14	Dias, N. (2017) [22]	7	Blanque, R. (2019) [32]	15
Cordero, R. (2017) [17]	14	Haakstad, L. (2015) [23]	8	Brik, M. (2018) [33]	5
Bacchi, M. (2017) [39]	8	Gustafsson, M.K. (2015) [24]	10	Coll, C. (2019) [12]	10
Haakstad, L. (2016) [4]	9	Charkamyani, F. (2019) [25]	5	Blanque, R. (2019) [34]	13
Sagedal, L.R. (2017) [18]	10	Perales, M. (2015) [26]	9	Awad, E. (2019) [35]	10
Sagedal, L.R. (2017) [19]	10	Palaez, M. (2019) [41]	8	Barakat, R. (2017) [5]	9
A-Cordero, M. (2018) [20]	15	Clark, E. (2019) [27]	8	Pawalia, A. (2017) [36]	7
Barakat, R. (2018) [40]	9	Haakstad, L. 2020) [28]	7	Barakat, R. (2016) [4]	9
Sanda, B. (2018) [37]	6	Blanque, R. (2020) [29]	15	Cordero, Y. (2015) [42]	9
Terrones, M. (2018) [21]	11	Blanque, R. (2020) [30]	15		
Blanque, R. (2017) [38]	14	Reoyo, O. (2019) [31]	6		

## Data Availability

Not applicable.

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
