# Peer review of "Can Group Exercise Programs Improve Health Outcomes in Pregnant Women? An Updated Systematic Review"

_ijerph, 2022, doi:10.3390/ijerph19084875_

Round 1

Reviewer 1 Report

First of all, thank you for the opportunity to review this article. it is a consistent, current and updated material, which brings added value to knowledge, on the targeted population segment. a few suggestions can be considered to increase the scientific value of the article.
some improvements are aimed at:
- use of acronyms (it is preferable to use the concept / concepts in complete form followed by the acronym (for example, RCT, PFMT, CSEP, etc.);
- references to sources in the text (see line 37 - year, line 76 - model 5A - World Health Organization, 2010, but also a study - 15. Carroll, JK, Fiscella, K., Epstein, RM et al. A 5A's communication intervention to promote physical activity in underserved populations, BMC Health Services Research, 2012, 12: 374, http://www.biomedcentral.com/1472-6963/12/374);
- sometimes ometimes the sentences are confusing in the forms (see lines 38-39, 43, 241 - NFFD - what studies ?, 333 - to add group exercises)
- the order of the authors in the titles: Sanchez-Garcia (see Table 3) is the first author, and in the bibliography appears the year 2019, and in the Table with the extracted data appears 2018, in References it is 2019
- some articles referred to do not refer to the effects of physical exercises performed in groups by subjects (pregnant women) - check [25, 33, 15].
- changing punctuation in the title (Can Group Exercise Programs Improve Health and Fitness Outcomes in Pregnant Women? An Updated Systematic Review)

- in the Introduction are developed, in general, the main effects on the target group - pregnant women, but without referring to the stage of knowing only the influences of physical exercises in the group.
- it would be useful to argue the choice of only the three databases in the selection of articles on the topic.
- to verify why the title included in [41] does not appear in the enumeration from subchapter 3.2.

Author Response

Response to REVIEWER 1

First of all, thank you for the opportunity to review this article. it is a consistent, current and updated material, which brings added value to knowledge, on the targeted population segment. a few suggestions can be considered to increase the scientific value of the article.

some improvements are aimed at:

Response: Thank you for the comment and the opportunity to improve the manuscript with your suggestions.

- use of acronyms (it is preferable to use the concept / concepts in complete form followed by the acronym (for example, RCT, PFMT, CSEP, etc.);

Response:

Pelvic Floor Muscle Training (PFMT) was already in line 55

…randomized control trials (RCT) [11]. Inserted in lines 75-76

…the Canadian Society for Exercise Physiology (CSEP) website. Inserted in lines 86-87

…described by the Preferred Reporting Items for Systematic Reviews and Meta-Analyses (PRISMA 2020). The PRISMA Statement. Inserted in lines 100-101

BMI - body mass index; CG – control group; FHR - fetal heart rate; GDM - gestational diabetes mellitus; GPAQ - global physical activity questionnaire; GWG – gestational weight gain; HC - hip circumference; HDL - high density lipoprotein; HRQoL - health-related quality of life; HRR - heart rate reserve; IDG - intervention diet group; IG – intervention group; LBP - low back pain; LDL - low density lipoprotein; MHR - maternal heart rate; OGTT - oral glucose tolerance test; PFM - pelvic floor muscles; PPD - postpartum depression; PW – pregnant women; PPWR - postpartum weight retention; WC - waist circumference. Inserted in lines 356-359

or the American College of Sports Medicine (ACSM) Inserted in line 425

- references to sources in the text (see line 37 - year, line 76 - model 5A - World Health Organization, 2010, but also a study - 15. Carroll, JK, Fiscella, K., Epstein, RM et al. A 5A's communication intervention to promote physical activity in underserved populations, BMC Health Services Research, 2012, 12: 374, http://www.biomedcentral.com/1472-6963/12/374);

Response:

Reference included: ACOG – American College of Obstetricians and Gynecologists. ACOG Committee Opinion No. 804: Physical Activity and Exercise During Pregnancy and the Postpartum Period. Obstetrics and Gynecology [Internet]. 2020; 135(4):[e178-e88 pp.].

- sometimes the sentences are confusing in the forms (see lines 38-39, 43, 241 NFFD - what studies ?, 333 - to add group exercises)

Response: corrected in the text and table

- the order of the authors in the titles: Sanchez-Garcia (see Table 3) is the first author, and in the bibliography appears the year 2019, and in the Table with the extracted data appears 2018, in References it is 2019

Response: corrected in the table

- some articles referred to do not refer to the effects of physical exercises performed in groups by subjects (pregnant women) - check [25, 33, 15].

Response:

15: Material and methods: A randomized clinical trial with 140 healthy pregnant women, aged between 21 and 43 years, divided into two groups, study (GE, n = 70) and control (GC, n = 70). The women were attracted at 12 weeks of gestation in the first trimester ultrasound control carried out in the different obstetrical services in Granada. They joined the program at week 20 of gestation and ended in week 37. The perinatal results were obtained from the partograph of each woman, included in the Delivery Room Services of the Complejo Hospitalario Universitario de Granada.

25: Participants in the aerobic exercise intervention (n = 14) completed three 50-min sessions weekly from 16 weeks gestation to delivery and were compared with a non-exercise control group (n = 22). 

33: The participants were randomly assigned into 2 equal groups: group A, undergoing an exercise program with a moderately restricted diet and insulin therapy, and group B (control group), receiving solely the same diet protocol with insulin therapy.

- changing punctuation in the title (Can Group Exercise Programs Improve Health and Fitness Outcomes in Pregnant Women? An Updated Systematic Review)

Response: The title has been changed according to the suggestion.

- in the Introduction are developed, in general, the main effects on the target group - pregnant women, but without referring to the stage of knowing only the influences of physical exercises in the group.

Response: in the introduction we describe the general benefits of physical activity and exercise in pregnancy outcomes and health and fitness parameters, as supported by the guidelines.

- it would be useful to argue the choice of only the three databases in the selection of articles on the topic.

Response: The choice of these three databases is based on the fact that they are the most recognized in the area of sports science, and free. PUBMED is a huge, reliable, and highly authoritative resource, specific to medicine and health.

- to verify why the title included in [41] does not appear in the enumeration from subchapter 3.2.

Response: 41 is a study protocol (in Spanish)

Reviewer 2 Report

The systematic review by de Castro et al., entitled Can Group Exercise Programs Improve Health and Fitness Outcomes in Pregnant Women: An Updated Systematic Review provides an update to the previously published systematic review (2015) on the topic of group prenatal exercise programs and maternal well-being. While I think this systematic review aims to address a specific question that is novel, I have some concerns that I feel should be addressed.

Introduction:

  1. There are many missing references in the introduction – the authors must support their statements made specifically with reference to the guidelines they mention (e.g. 30-60minutes of duration per session is not in any guideline I am familiar with).
  2. To much of the introduction is a summary of the guidelines and a comparison of guidelines rather than a concise explanation of pertinent literature to drive the reader towards the research question. I would suggest shortening this section.
  3. The review in reference #2 (Evenson et al 2014) is outdated and therefore any information obtained from is not necessarily valid as the Canadian and American guidelines have both been updated since that was published.
  4. Similarly, paragraph from line 65-69 is somewhat contradictory and could be incorporated within.
  5. Do you think group exercise with mixed resistance, aerobic and pelvic floor muscle training would be superior to “no exercise” or other control condition? What is your rationale to study only group exercise for this review?

Methods:

  1. The authors have updated their original search to include data from 2015-2020. Why was this not updated to include more current research? This is being submitted in March of 2022 and should be updated to include more recent data if it exists. This reviewer is familiar with the amount of work required to do this, but strongly recommends an updated literature search and inclusion of any recent evidence prior to publication.
  2. Regarding the “types of interventions” section of the methods: No mention of group exercise (your research question!).
  3. Why did interventions need to be at least 12 weeks in duration?
  4. Why were studies designed specifically for overweight & obese individuals excluded (94 studies!!!)? We know these groups benefit the most from prenatal exercise.
  5. What is your control group or comparator?
  6. Outcome measures are vague – please state all outcomes of interest explicitly. This appears in the results (line 261). Please move to methods. Please also include information about why you choose these specific outcomes in the introduction.
  7. Why is fitness (e.g. VO2) not an outcome of interest? It is in your title.
  8. How did the authors deal with studies which reported data from the same women in more than one publication?
  9. Regarding “data extraction” section: how was exercise intensity defined?

Results:

  1. Line 155, remove the word “just”
  2. Why was the data from this search (2015-2020) not combined with the previous data (i.e., prior to 2015)? This is therefore not an update but rather a very limited view on this topic.
  3. Line 228: please define SWEP.
  4. Line 244: please define NFFD

Discussion:

  1. I again ask the authors to provide evidence and rationale for studying group exercise only and put their results from studies published exclusively between 2015-2020 into the context of the previous systematic review on this topic and also the general literature.
  2. I recommend the authors to read the systematic reviews/ meta-analysis that informed the current Canadian and American (or other international) guidelines as many of these outcomes have clearly been addressed by these prior reviews and the novelty of the research presented here is becoming lost to me the way it is presented.
  3. In general, more references outside of your 31 manuscripts you are reviewing is needed for the discussion. Please do not simply re-tell your results.
  4. Not all outcomes of interest (results line 260) are discussed. Why not?
  5. Paragraphs starting at line 520 and 524 seems to contradict one another.

Minor comments from discussion:

  1. Line 346 – this is contradictory seeing as you purposefully excluded all research on this topic.
  2. The sections on GDM and gestational hypertension are extremely flawed based on your study selection (exclusion of overweight and obese).
  3. Line 375: You are not qualified to make this recommendation based on your data, either cite the literature that makes this recommendation or remove.
  4. Line 378: this is untrue, we know that prenatal physical activity reduces the odds of developing PE by ~40%. Please see PMID: 30337463.
  5. Line 392-395 – this is great and what I have been looing for in your discussion, thank you!
  6. Section 4.5, 4.7 – please state how does this relate to group exercise in normal weight individuals?
  7. Line 530: maternal blood pressure was not an outcome measure of interest, please remove.
  8. Line 552 – missing information here (currently highlighted yellow?)

Abstract:

I have this section last, as I read this last and made these comments after reading the rest of the paper - please make adjustments within the paper accordingly as well as within the abstract.

  1. Line 21 – I don’t think this is well characterized in the manuscript- perhaps do not include in the abstract.
  2. Line 27-29: this conclusion is not well supported by your results and should not be stated this way.

Author Response

Response to REVIEWER 2

The systematic review by de Castro et al., entitled Can Group Exercise Programs Improve Health and Fitness Outcomes in Pregnant Women: An Updated Systematic Review provides an update to the previously published systematic review (2015) on the topic of group prenatal exercise programs and maternal well-being. While I think this systematic review aims to address a specific question that is novel, I have some concerns that I feel should be addressed.

Response: Thank you for the comment, the extensive revision, and the opportunity to improve the manuscript with your suggestions.

Introduction:

  1. There are many missing references in the introduction – the authors must support their statements made specifically with reference to the guidelines they mention (e.g. 30-60minutes of duration per session is not in any guideline I am familiar with).

Response: ACOG 2020 reference inserted:

  1. To much of the introduction is a summary of the guidelines and a comparison of guidelines rather than a concise explanation of pertinent literature to drive the reader towards the research question. I would suggest shortening this section.

Response:

Text removed: Combining aerobic exercise and resistance training during pregnancy is more effective at improving health outcomes than interventions solely focused on aerobic exercise. It is important to note that, for some outcomes, lower-intensity physical activity also imparts benefits, although accumulating more physical activity over the week was associated with greater benefits

Text removed: Obstetric care and exercise professionals must carefully consider the potential costs and perceived barriers to prenatal physical activity to facilitate participation. The guidelines suggest the use of The Five A’s (Ask, Advise, Assess, Assist, and Arrange) to promote consistent exercise practice. Monitoring a pregnancy intervention requires a multidisciplinary team, which not only includes health professionals but also exercise physiologists. Hence, it is necessary that the language, ideas, and recommendations be clear and consensual among the various stakeholders [3].

  1. The review in reference #2 (Evenson et al 2014) is outdated and therefore any information obtained from is not necessarily valid as the Canadian and American guidelines have both been updated since that was published.

Response:

The paragraph was rewritten: In 2014, Evenson et al. compared pregnancy-related physical activity guidelines from around the world [2]. The authors suggested that there should be more consensus between the guidelines to enable proper cooperation between health professionals, exercise professionals, and pregnant women [2]. This review supported that the creation of a common language to facilitate the communication between the various stakeholders, may help increase the adherence to the physical activity guidelines by both the providers and their patients. More recently, these authors fostered that the 4 recent guidelines can facilitate use of updated recommendations by health care providers regarding physical activity during pregnancy [REF]. Furthermore, to facilitate the use of guidelines in practice, there are other tools available at the Canadian Society for Exercise Physiology (CSEP) website.

New reference was included: Evenson KR, Mottola MF, Artal R. Review of Recent Physical Activity Guidelines During Pregnancy to Facilitate Advice by Health Care Providers. Obstet Gynecol Surv. 2019 Aug;74(8):481-489. doi: 10.1097/OGX.0000000000000693. PMID: 31418450.

  1. Similarly, paragraph from line 65-69 is somewhat contradictory and could be incorporated within.

Response:

Text removed: Comparing and updating the guidelines, ACOG [1], contrary to the Canadian Guidelines [3], states that the contraindications for exercise (absolute and relative) should only be considered in aerobic exercise. Thus, a pregnant woman with obstetric or medical complications should maintain strength or pelvic floor training. Furthermore, ACOG in 2015 reported that the evidence of the benefits of aerobic training was limited. However, the same article was recently revised and redacted this fact (2020) [1].

  1. Do you think group exercise with mixed resistance, aerobic and pelvic floor muscle training would be superior to “no exercise” or other control condition? What is your rationale to study only group exercise for this review?

Response:

Two of the researchers of this paper have a long professional experience in working with pregnant women in group and individually. Based on practice, these authors are deeply convinced that group exercise with mixed resistance, aerobic and pelvic floor muscle training (and other types of exercises) are pretty much superior to “no exercise” or other control condition.

We decided to limit our review to group interventions, bearing in mind that the exercise adherence may be substantially influenced by the group interaction (1). In the study by Blackstone et al. (2) women underlined the importance of the social aspect of group classes in motivating participants to exercise. During pregnancy these aspects may play key role in exercise adherence, translating into the effectiveness and health benefits of exercise programs.

  1. Spink KS, Carron AV. Group Cohesion and Adherence in Exercise Classes. Journal of Sport & Exercise Psychology. 1992;14(1):78-86.
  2. Blackstone SR, Reeves D, Lizzo R, Graber KC. A QUALITATIVE INQUIRY OF MOTIVATIONS TO PARTICIPATE IN GROUP EXERCISE AMONG WOMEN. American Journal of Health Studies. 2017;32(2):78-89.

Cristina Jorge, Rita Santos-Rocha and Teresa Bento. Can Group Exercise Programs Improve Health Outcomes in Pregnant Women? A Systematic Review

Volume 11 , Issue 1 , 2015, Page: [75 - 87]Pages: 13

DOI: 10.2174/157340481101150914202014

The present paper is an update.

Methods:

  1. The authors have updated their original search to include data from 2015-2020. Why was this not updated to include more current research? This is being submitted in March of 2022 and should be updated to include more recent data if it exists. This reviewer is familiar with the amount of work required to do this, but strongly recommends an updated literature search and inclusion of any recent evidence prior to publication.

Response:

As shown in https://pubmed.ncbi.nlm.nih.gov/?term=exercise%20and%20pregnancy&filter=pubt.randomizedcontrolledtrial&filter=datesearch.y_1&filter=hum_ani.humans&filter=sex.female

There is no recent data in 2021, regarding the impact of in-person group exercise on maternal health and fitness parameters, possibly due to the 2020 pandemic of COVID-19. Due to the COVID-19 pandemic since 2020, and in accordance with national and international ACSM Exercise is Medicine recommendations [ACSM. COVID-19 and Exercise. Exercise is Medicine. Available at: https://www.exerciseismedicine.org/eim-in-action/health-care/resources/covid-19-and-exercise1/ ] that advised pregnant women to avoid indoor gyms, and recommends limiting social gatherings, a series of exercise proposals to be active at home were published. Moreover, since the COVID-19 pandemic caused fitness facilities to close and restructure services, the first trends became wearable technology and online training (i.e., developed for the at-home exercise experience, this trend uses digital streaming technology to deliver group, individual or instructional exercise programs online).

Unfortunately, it caused adverse effects in pregnant women (e.g. Lebel C, MacKinnon A, Bagshawe M, Tomfohr-Madsen L, Giesbrecht G. Elevated depression and anxiety symptoms among pregnant individuals during the COVID-19 pandemic. J Affect Disord. 2020 Dec 1;277:5-13. doi: 10.1016/j.jad.2020.07.126. Epub 2020 Aug 1. Erratum in: J Affect Disord. 2021 Jan 15;279:377-379. PMID: 32777604; PMCID: PMC7395614.) and (Ayaz R, HocaoÄŸlu M, Günay T, Yardımcı OD, Turgut A, Karateke A. Anxiety and depression symptoms in the same pregnant women before and during the COVID-19 pandemic. J Perinat Med. 2020 Nov 26;48(9):965-970. doi: 10.1515/jpm-2020-0380. PMID: 32887191.)

  1. Regarding the “types of interventions” section of the methods: No mention of group exercise (your research question!).

Response: corrected.

The articles included in this review were based in prenatal in-person group exercise interventions with a minimum duration of 12 weeks, supervised by physical exercise specialists or exercise physiologists.

  1. Why did interventions need to be at least 12 weeks in duration?

Response:

12 to 16 weeks of duration,  is a criterion of comparison in most studies with different populations, and the amount of time required to achieve significant improvement from any training program, as shown in https://pubmed.ncbi.nlm.nih.gov/?term=12-week+exercise+effectiveness&filter=pubt.randomizedcontrolledtrial&filter=datesearch.y_5&filter=hum_ani.humans&filter=sex.female

  1. Why were studies designed specifically for overweight & obese individuals excluded (94 studies!!!)? We know these groups benefit the most from prenatal exercise.

Response: Although we recognize dwell on the role of the exercise in this specific group, the objective of this study was to update a previous systematic review. For this reason, the inclusion and exclusion criteria were maintained.

Moreover, at least 3 recent systematic reviews including overweight and obese pregnant women, were published:

Du MC, Ouyang YQ, Nie XF, Huang Y, Redding SR. Effects of physical exercise during pregnancy on maternal and infant outcomes in overweight and obese pregnant women: A meta-analysis. Birth. 2019 Jun;46(2):211-221. doi: 10.1111/birt.12396. Epub 2018 Sep 21. PMID: 30240042.

Flannery C, Fredrix M, Olander EK, McAuliffe FM, Byrne M, Kearney PM. Effectiveness of physical activity interventions for overweight and obesity during pregnancy: a systematic review of the content of behaviour change interventions. Int J Behav Nutr Phys Act. 2019 Nov 1;16(1):97. doi: 10.1186/s12966-019-0859-5. PMID: 31675954; PMCID: PMC6825353.

Simon A, Pratt M, Hutton B, Skidmore B, Fakhraei R, Rybak N, Corsi DJ, Walker M, Velez MP, Smith GN, Gaudet LM. Guidelines for the management of pregnant women with obesity: A systematic review. Obes Rev. 2020 Mar;21(3):e12972. doi: 10.1111/obr.12972. Epub 2020 Jan 14. PMID: 31943650; PMCID: PMC7064940.

  1. What is your control group or comparator?

Response: 1) no exercise control group (i.e., receiving “standard care“, or included in a “waiting list”); 2) no exercise but with other type of intervention group (e.g., motivational counselling); 3) control group (i.e., intervention group 2) engaged in a different exercise program (e.g., supervised vs non supervised program, group vs personal training, multicomponent vs not mixed exercise, etc.). Thank you for this question. This information was included in section 2.1.3.

  1. Outcome measures are vague – please state all outcomes of interest explicitly. This appears in the results (line 261). Please move to methods. Please also include information about why you choose these specific outcomes in the introduction.

Response:

Thank you for this question. This information was included in section 2.1.4.

Considering the aims of this systematic review, the main outcome variables in analysis are related to maternal health (i.e., quality of life, weight gain, gestational diabetes, cholesterol, hypertension, pre-eclampsia, well-being, depression, rest heart rate, sleep quality, low back pain), maternal physical activity (i.e., level and type of physical activity), and maternal fitness (i.e., cardiorespiratory fitness, strength, flexibility, balance, coordination, pelvic floor muscle strength), and labor outcomes (i.e., mode of delivery, birth weight, fetal cardiac function, and Apgar score)

  1. Why is fitness (e.g. VO2) not an outcome of interest? It is in your title.

Response:

This information was included in section 2.1.4.

Considering the aims of this systematic review, the main outcome variables in analysis are related to … maternal fitness (i.e., cardiorespiratory fitness, strength, flexibility, balance, coordination, pelvic floor muscle strength), and …

  1. How did the authors deal with studies which reported data from the same women in more than one publication?

Response: some studies reported data from the same women in more than one publication, but targeting different outcome variables.

  1. Regarding “data extraction” section: how was exercise intensity defined?

Response: the intensity of exercise is defined either using the Borg scale or the percentage of heart rate reserve, as shown, for instances, in line 277: The intensity of the intervention was measured with Borg’s scale remaining between 12 and 14, or with heart rate remaining at maximum values of 60% [8,31,29] or 70% [7]. Some studies used heart rate with women who reported a value higher than 14 on Borg’s Scale [27]….

Results:

  1. Line 155, remove the word “just”

Response: done.

  1. Why was the data from this search (2015-2020) not combined with the previous data (i.e., prior to 2015)? This is therefore not an update but rather a very limited view on this topic.

Response: in the previous review, a total of 17 studies were selected for analysis. All studies were randomized control trials conducted with pregnant women that evaluated the effect of group exercise programs on the health outcomes of mother and newborn. Studies were conducted between 2009 and 2014, based on the ACOG guidelines published in 2002. New guidelines were published in 2015, affecting clinical, research and exercise practices regarding physical activity during pregnancy. These are the reasons for not including trials conducted prior to 2015.

  1. Line 228: please define SWEP.

Response: SWEP = Study Water Exercise Pregnant, as defined by Aguilar-Cordero MJ. Influencia del programa SWEP (Study Water Exercise Pregnant) en los resultados perinatales: protocolo de estudio. Nutr Hosp. 2015. doi:10.20960/nh.02870

It was removed from text.

  1. Line 244: please define NFFD

Response: NFFD = the Norwegian Fit for Delivery, as defined by

Sagedal LR, Øverby NC, Bere E, et al. Lifestyle intervention to limit gestational weight gain: the Norwegian Fit for Delivery randomised controlled trial. BJOG An Int J Obstet Gynaecol. 2017;124(1):97-109. doi:10.1111/1471-0528.13862

And

Sagedal LR, Vistad I, Øverby NC, et al. The effect of a prenatal lifestyle intervention on glucose metabolism: Results of the Norwegian Fit for Delivery randomized controlled trial. BMC Pregnancy Childbirth. 2017;17(1):1-12. doi:10.1186/s12884-017-1340-6

It was removed from text.

Discussion:

  1. I again ask the authors to provide evidence and rationale for studying group exercise only and put their results from studies published exclusively between 2015-2020 into the context of the previous systematic review on this topic and also the general literature.

Response: done (second paragraph of the discussion).

  1. I recommend the authors to read the systematic reviews/ meta-analysis that informed the current Canadian and American (or other international) guidelines as many of these outcomes have clearly been addressed by these prior reviews and the novelty of the research presented here is becoming lost to me the way it is presented.

Response:

The authors are deeply aware of all SR and guidelines related with the topic.

Moreover, A recent review identified important gaps in research, including a lack of studies investigating the benefits of group interventions (Di Lorito C, Long A, Byrne A, Harwood RH, Gladman JRF, Schneider S, Logan P, Bosco A, van der Wardt V. Exercise interventions for older adults: A systematic review of meta-analyses. J Sport Health Sci. 2021 Jan;10(1):29-47. doi: 10.1016/j.jshs.2020.06.003. Epub 2020 Jun 7. PMID: 32525097; PMCID: PMC7858023.). The 12 systematic reviews/meta-analysis produced by the research group involved in the current 2019 Canadian guidelines, address all types of physical activity and exercise programs. The novelty of this SR is precisely the inclusion of group exercise programs, and the use of CERT to assess those programs. As far as we are concerned, there are no studies investigating the benefits of group interventions with pregnant women. We hope that the corrections and improvements made at this stage, will contribute to clarify this point.

  1. In general, more references outside of your 31 manuscripts you are reviewing is needed for the discussion. Please do not simply re-tell your results.

Response: done. We hope that the corrections made in section 4, will contribute to improve this point.

  1. Not all outcomes of interest (results line 260) are discussed. Why not?

Response: Not all outcomes of interest were addressed in the studies included in the analysis.

  1. Paragraphs starting at line 520 and 524 seems to contradict one another.

Response: corrected (lines 712).

Minor comments from discussion:

  1. Line 346 – this is contradictory seeing as you purposefully excluded all research on this topic.

Response: section 4.1 was revised.

  1. The sections on GDM and gestational hypertension are extremely flawed based on your study selection (exclusion of overweight and obese).

Response: GDM and gestational hypertension may occur in pregnant women with normal weight gain.

  1. Line 375: You are not qualified to make this recommendation based on your data, either cite the literature that makes this recommendation or remove.

Response: section 4.2 was revised.

  1. Line 378: this is untrue, we know that prenatal physical activity reduces the odds of developing PE by ~40%. Please see PMID: 30337463.

Response: the word “unclear” was a mistake. Two references were inserted:

  1. Magro-Malosso ER, Saccone G, Di Tommaso M, Roman A, Berghella V. Exercise during pregnancy and risk of gestational hypertensive disorders: a systematic review and meta-analysis. Acta Obstet Gynecol Scand. 2017 Aug;96(8):921-31.
  2. Davenport MH, Ruchat SM, Poitras VJ, Jaramillo Garcia A, Gray CE, Barrowman N, Skow RJ, Meah VL, Riske L, Sobierajski F, James M, Kathol AJ, Nuspl M, Marchand AA, Nagpal TS, Slater LG, Weeks A, Adamo KB, Davies GA, Barakat R, Mottola MF. Prenatal exercise for the prevention of gestational diabetes mellitus and hypertensive disorders of pregnancy: a systematic review and meta-analysis. Br J Sports Med. 2018 Nov;52(21):1367-75.

  1. Line 392-395 – this is great and what I have been looing for in your discussion, thank you!

Response: thank you for this comment.

  1. Section 4.5, 4.7 – please state how does this relate to group exercise in normal weight individuals?

Response: done.

4.5 Reference inserted: Davenport MH, Ruchat SM, Sobierajski F, Poitras VJ, Gray CE, Yoo C, Skow RJ, Jaramillo Garcia A, Barrowman N, Meah VL, Nagpal TS, Riske L, James M, Nuspl M, Weeks A, Marchand AA, Slater LG, Adamo KB, Davies GA, Barakat R, Mottola MF. Impact of prenatal exercise on maternal harms, labour and delivery outcomes: a systematic review and meta-analysis. Br J Sports Med. 2019 Jan;53(2):99-107. doi: 10.1136/bjsports-2018-099821. Epub 2018 Oct 18. PMID: 30337349.

4.7 Reference inserted: Davenport MH, Meah VL, Ruchat SM, Davies GA, Skow RJ, Barrowman N, Adamo KB, Poitras VJ, Gray CE, Jaramillo Garcia A, Sobierajski F, Riske L, James M, Kathol AJ, Nuspl M, Marchand AA, Nagpal TS, Slater LG, Weeks A, Barakat R, Mottola MF. Impact of prenatal exercise on neonatal and childhood outcomes: a systematic review and meta-analysis. Br J Sports Med. 2018 Nov;52(21):1386-1396. doi: 10.1136/bjsports-2018-099836. PMID: 30337465.

  1. Line 530: maternal blood pressure was not an outcome measure of interest, please remove.

Response: maternal blood pressure was an outcome measure of interest.

  1. Line 552 – missing information here (currently highlighted yellow?)

Response: yes. We will provide the reference number, after acceptance of the article.

Abstract:

I have this section last, as I read this last and made these comments after reading the rest of the paper - please make adjustments within the paper accordingly as well as within the abstract.

  1. Line 21 – I don’t think this is well characterized in the manuscript- perhaps do not include in the abstract.

Response: corrected.

  1. Line 27-29: this conclusion is not well supported by your results and should not be stated this way.

Response: corrected.

Reviewer 3 Report

Review of " Can Group Exercise Programs Improve Health and Fitness Outcomes in Pregnant Women: An Updated Systematic Review"

This is an exceptional review of the literature. The authors have conducted an extensive review and compiled results of various studies to present in a very succinct and logical way. I can only imagine this report receiving numerous citations and  being a classic review for many health care professionals. The 27-item checklist and a four-phase flow diagram demonstrates the seriousness of this review of the literature. Carefully reviewing and critiquing the published studies in how the studies were conducted and presented appears to be unbiased and clearly presented. Personally, I am very impressed on the efforts but into this review.

I do not see any edits required for this manuscript.

I do think if the authors were interested in another study/review on a similar topic, which would likely be of broad interest, is on the topic of ethnic (cultural) views of exercise during pregnancy. Differences exists in various cultures in what women shouldn't do or allowed to do while pregnant.  

Author Response

Response to REVIEWER 3

This is an exceptional review of the literature. The authors have conducted an extensive review and compiled results of various studies to present in a very succinct and logical way. I can only imagine this report receiving numerous citations and  being a classic review for many health care professionals. The 27-item checklist and a four-phase flow diagram demonstrates the seriousness of this review of the literature. Carefully reviewing and critiquing the published studies in how the studies were conducted and presented appears to be unbiased and clearly presented. Personally, I am very impressed on the efforts but into this review.

I do not see any edits required for this manuscript.

I do think if the authors were interested in another study/review on a similar topic, which would likely be of broad interest, is on the topic of ethnic (cultural) views of exercise during pregnancy. Differences exists in various cultures in what women shouldn't do or allowed to do while pregnant. 

Response: Thank you so much for your appreciation of the document. We appreciate the comments and compliments to the article. We also appreciate the suggestion of future study that we can take into account.

Round 2

Reviewer 2 Report

Reply:

Thank you to the authors for a quick turn-around and reply to my initial comments. I do feel that many replies are quite defensive with minimal change to the manuscript. I would like to let the authors know that I carefully reviewed the manuscript as this topic is very important to me and all of my comments are made to improve this work.

I will note that I have also read the reviews and replies of the other reviewer. However – I am just one person judging this work & our peers can decide for themselves their opinions of the work after its published– so I will only comment on major things at this point that I think are required for this to be published.

  1. I am still not convinced that the question being asked (effects of group exercise during pregnancy versus “anything else” on a pile of unrelated outcomes) is well supported by the introduction. For example your reply that the authors have “long professional experience in working with pregnant women in group and individually. Based on practice, these authors are deeply convinced that group exercise with mixed resistance, aerobic and pelvic floor muscle training (and other types of exercises) are pretty much superior to “no exercise” or other control condition” is not scientific rationale for asking this question. Your outcomes of interest need to be addressed by the introduction as well.
  2. Lines 41-49 still need references for these claims. The ACOG 2020 reference is not sufficient for each of your statements here. I gave the specific example of your comment regarding 30-60minutes per session not being supported by either guideline.
  3. In the methods – if there are no new publications after 2020 as you state I your reply to reviewers, please state that your search was updated to confirm this. If you didn’t find anything new – that’s fine, but unless you looked you cannot say there was no research. Covid aside, many publications from prior studies in this area were published the past two years.
  4. I had previously asked why fitness is in your title, you did indicate that it is an outcome of interest- but it appears to be one with no information in the literature. Therefore, I recommend updating the title as your manuscript provides no insight on this topic (and please add a section in the discussion regarding this topic as it is an important avenue for future research).
  5. My initial comment “please provide evidence and rationale for studying group exercise only and put their results from studies published exclusively between 2015-2020 into the context of the previous systematic review on this topic and also the general literature.” Was not well addressed. The authors DID provide rational for the timeframe for their search (thank you!) but have not put this data in context with previous reviewer by this group or the other literature. What I really want to know is: Did the findings of this systematic review support group exercise, for which outcomes, why or why not?

I look forward to reading your reply and updated manuscript.

Author Response

Dear Reviewer,

Thank you once more for the comment and the opportunity to improve the manuscript with your suggestions.

Reviewer: Thank you to the authors for a quick turn-around and reply to my initial comments. I do feel that many replies are quite defensive with minimal change to the manuscript. I would like to let the authors know that I carefully reviewed the manuscript as this topic is very important to me and all of my comments are made to improve this work.

Response: This topic is also very important to us, and we put a lot of effort on it. We pretty much appreciate your time and feedback to improve this work.

Reviewer: I will note that I have also read the reviews and replies of the other reviewer. However – I am just one person judging this work & our peers can decide for themselves their opinions of the work after its published– so I will only comment on major things at this point that I think are required for this to be published.

  1. I am still not convinced that the question being asked (effects of group exercise during pregnancy versus “anything else” on a pile of unrelated outcomes) is well supported by the introduction. For example your reply that the authors have “long professional experience in working with pregnant women in group and individually. Based on practice, these authors are deeply convinced that group exercise with mixed resistance, aerobic and pelvic floor muscle training (and other types of exercises) are pretty much superior to “no exercise” or other control condition” is not scientific rationale for asking this question. Your outcomes of interest need to be addressed by the introduction as well.

Response: Recent systematic reviews have been showing the effectiveness of physical activity in several maternal health outcomes and in a few fitness parameters [11-14, 24, 25]. However, it is not clear which features are included in the most effective exercise programs, regarding outcomes and long-term adherence.

We have improved the introduction and the discussion section. So, we hope it is more clear now.

Reviewer:

2. Lines 41-49 still need references for these claims. The ACOG 2020 reference is not sufficient for each of your statements here. I gave the specific example of your comment regarding 30-60minutes per session not being supported by either guideline.

Response: We provided feedback to your comment regarding 30-60 minutes per session being supported by ACOG 2020 guideline.

In the introduction section: “…These guidelines (ACOG) suggest that pregnant women without contraindications should accumulate at least 150 min/week of moderate-intensity aerobic exercise spread across, at least, 3 days per week, in sessions of 30-60 minutes [5].”

Reviewer:

3. In the methods – if there are no new publications after 2020 as you state I your reply to reviewers, please state that your search was updated to confirm this. If you didn’t find anything new – that’s fine, but unless you looked you cannot say there was no research. Covid aside, many publications from prior studies in this area were published the past two years.

Response: We searched but did not find new publications after 2020 regarding (in-person) RCT testing the effectiveness of prenatal exercise programs. Probably due to the pandemic, online classes were the main context of intervention. Moreover, “Pregnant women are receptive to online group exercise classes and expressed that they are an accessible option to accommodating physical activity during the pandemic”, as reported by: Silva-Jose, C., Nagpal, T.S., Coterón, J. et al. The ‘new normal’ includes online prenatal exercise: exploring pregnant women’s experiences during the pandemic and the role of virtual group fitness on maternal mental health. BMC Pregnancy Childbirth 22, 251 (2022). https://doi.org/10.1186/s12884-022-04587-1

Reviewer:

4. I had previously asked why fitness is in your title, you did indicate that it is an outcome of interest- but it appears to be one with no information in the literature. Therefore, I recommend updating the title as your manuscript provides no insight on this topic (and please add a section in the discussion regarding this topic as it is an important avenue for future research).

Response: We agree with this suggestion. The title was updated, as well as the discussion section.

Reviewer:

5. My initial comment “please provide evidence and rationale for studying group exercise only and put their results from studies published exclusively between 2015-2020 into the context of the previous systematic review on this topic and also the general literature.” Was not well addressed. The authors DID provide rational for the timeframe for their search (thank you!) but have not put this data in context with previous reviewer by this group or the other literature. What I really want to know is: Did the findings of this systematic review support group exercise, for which outcomes, why or why not?

Response: Yes, they did. The discussion section was updated in order to make it more clear.

Group exercise is a popular form of physical activity, and continues to be one of the fitness trends, despite the pandemic:

Thompson W. Worldwide Survey of Fitness Trends for 2021. ACSM's Health & Fitness Journal: 1/2 2021 - Volume 25 - Issue 1 - p 10-19. doi: 10.1249/FIT.0000000000000631

Thompson W. Worldwide Survey of Fitness Trends for 2022. ACSM's Health & Fitness Journal: 1/2 2022 - Volume 26 - Issue 1 - p 11-20. doi: 10.1249/FIT.0000000000000732

Reviewer: I look forward to reading your reply and updated manuscript.

Response:

We look forward to properly addressing your suggestions.

Thank you very much for your feedback.

Round 3

Reviewer 2 Report

no further comment